# Dynamic Priors in Bayesian Optimization for Hyperparameter Optimization

## Abstract

Hyperparameter optimization (HPO), for example, based on Bayesian optimization (BO), supports users in designing models well-suited for a given dataset. HPO has proven its effectiveness on several applications, ranging from classical machine learning for tabular data to deep neural networks for computer vision and transformers for natural language processing. However, HPO still sometimes lacks acceptance by machine learning experts due to its black-box nature and limited user control. Addressing this, first approaches have been proposed to initialize BO methods with expert knowledge. However, these approaches do not allow for online steering *during* the optimization process. In this paper, we introduce a novel method that enables repeated interventions to steer BO via user input, specifying expert knowledge and user preferences at runtime of the HPO process in the form of prior distributions. To this end, we generalize an existing method, $\pi$BO, preserving theoretical guarantees. We also introduce a misleading prior detection scheme, which allows protection against harmful user inputs. In our experimental evaluation, we demonstrate that our method can effectively incorporate multiple priors, leveraging informative priors, whereas misleading priors are reliably rejected or overcome. Thereby, we achieve competitiveness to unperturbed BO.

## 1 Introduction

Hyperparameter optimization (HPO) is concerned with automatically optimizing the hyperparameters of machine learning algorithms for a given task. A task often consists of a dataset, a configuration space, and a performance measure (Hutter et al., 2019). HPO is effective on classical machine learning (Eggensperger et al., 2021; Bansal et al., 2022; Pfisterer et al., 2022) as well as on deep neural networks and transformers for computer-vision and natural language processing (Müller et al., 2023; Wang et al., 2024; Rakotoarison et al., 2024; Pineda Arango et al., 2024). Nevertheless, users often disregard HPO in favor of maintaining manual hyperparameter configuration control (Bouthillier & Varoquaux, 2020; Van der Blom et al., 2021; Kannengießer et al., 2025). The reluctance is amplified in the case of knowledgeable experts (Kannengießer et al., 2025). At the same time, explainability methods for HPO (Wang et al., 2019; Sass et al., 2022; Zöller et al., 2023) seek to provide users with insights into the optimization process, often motivating adaptations to HPO. However, approaches that allow practitioners to steer the optimization process, instead of restarting it, remain largely underexplored. Nevertheless, they could tackle expert users' perceived lack of control (Kannengießer et al., 2025), thus possibly furthering HPO methods' acceptance.

Recently, initial steps have been taken toward collaborative automated machine learning (AutoML) approaches, subsuming HPO, in view of a more human-centered AutoML paradigm (Lindauer et al., 2024). More specifically, Hvarfner et al. (2022) and Mallik et al. (2023) propose user-centric interfaces for Bayesian optimization (BO) (Jones et al., 1998) and Hyperband (Li et al., 2017), respectively. Using explicit user priors on the location of the optimal configuration, the interfaces enable users to bias the optimization process. Hvarfner et al. (2024) extend this idea by building a general framework that enables users to encode properties of the optimized function, such as the maximum achievable performance, into the optimization process. These approaches show that informative priors can significantly enhance performance, while misleading priors do not break the optimization process, and the associated performance deterioration can be recovered from in the long run. Although these developments represent progress toward human-centered BO, the proposed level of interaction remains limited, as the approaches lack the possibility of user-enabled online control.

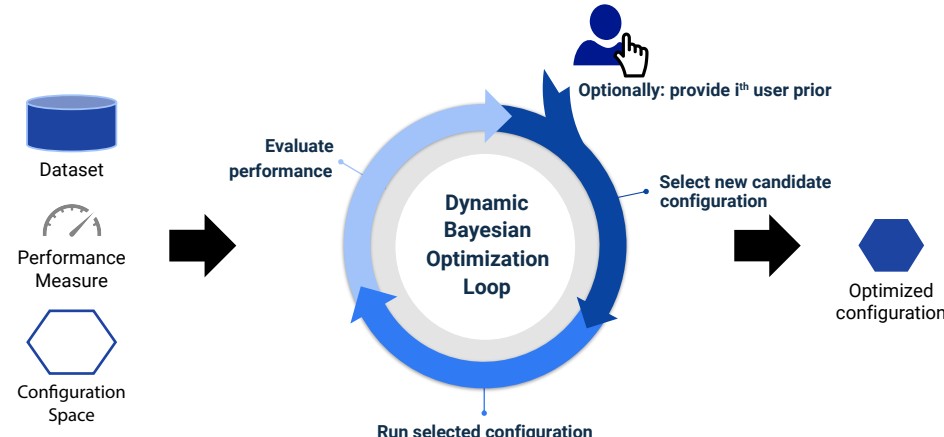

Figure 1: Overview of the proposed dynamic Bayesian optimization (`DynaBO`) method. Provided a dataset, a performance measure, a configuration space, and an optional initial prior, the loop iteratively selects new hyperparameter configurations. At each step, a candidate configuration is evaluated, and it is assessed. The process continues until an optimized configuration is identified. The framework allows users to steer the optimization process by dynamically adding priors at runtime.

In this paper, we propose an approach for dynamic Bayesian optimization, dubbed `DynaBO`, to advance the interactivity of BO-based HPO by enabling, but not requiring, users to inject knowledge on potentially well-suited regions of the hyperparameter configuration space. This user knowledge is provided in the form of user priors and can be injected at any point during the process, as illustrated in Figure 1. When multiple priors are supplied simultaneously or in close succession, they are stacked and their effects are combined. This (i) allows users to continuously steer the optimization process, and (ii) enables them to use HPO in the typical rapid-prototyping workflows of machine learning practitioners (Studer et al., 2021) where users oversee and direct the construction of the continuously updated model. These advantages are enhanced by an informed user, e.g., through utilizing explainability methods for HPO (Hutter et al., 2014; Moosbauer et al., 2021; Segel et al., 2023), which provide actionable insights.

Concretely, `DynaBO` generalizes the work of Hvarfner et al. (2022) from a single to multiple priors provided by users over time. Importantly, `DynaBO` maintains $\pi$BO's speedup for informative priors, and retains its theoretical convergence properties despite misleading priors. To tackle the nevertheless significant anytime performance deterioration due to misleading priors, we propose an additional mechanism for their detection and potential rejection. This mechanism is based on an assessment of the provided priors using acquisition functions and the already trained surrogate model, therefore requiring no significant overhead. Our experimental evaluation on a multitude of real-world HPO tasks reveals substantial speedups of `DynaBO` over vanilla BO and $\pi$BO (Hvarfner et al., 2022) for informative priors. At the same time, while $\pi$BO suffers from performance deterioration due to a single misleading prior, our prior rejection mechanism enables `DynaBO` to achieve competitiveness with vanilla BO even when exposed to misleading priors repeatedly.

## 2 HYPERPARAMETER OPTIMIZATION

Hyperparameter optimization (HPO) is concerned with finding a suitable hyperparameter configuration of a learner for a given task, typically defined by a labeled dataset $D$ and a loss function $\ell$ that evaluates predictive quality (Bischl et al., 2023). Given an input space $\mathcal{X}$ and an output (label) space $\mathcal{Y}$, $D = \{(x^{(k)}, y^{(k)})\}_{k=1}^{N}$ is a finite sample drawn from a joint probability distribution $P(\mathcal{X}, \mathcal{Y})$. $D$ is split into a training set $D_T$ and validation set $D_V$. We consider a loss function $\ell : \mathcal{Y} \times P(\mathcal{Y}) \rightarrow \mathbb{R}$ that evaluates the quality of predictions in terms of the true label and a probability distribution over $\mathcal{Y}$. Given a hyperparameter configuration $\lambda \in \Lambda$ from a configuration space $\Lambda$, and dataset $D_T$, the learner produces a hypothesis $h_{\lambda, D_T} \in \mathcal{H} := \{h \mid h : \mathcal{X} \rightarrow P(\mathcal{Y})\}$, that is, a mapping from inputs to probability distributions over $\mathcal{Y}$.

Since the hyperparameter configuration $\lambda$ influences both the hypothesis space $\mathcal{H}$ and the inductive behavior of the learner, selecting a well-suited hyperparameter configuration with respect to the dataset and the loss function is key to achieving peak performance. The objective of HPO is to find a configuration $\lambda^*$ that leads to a hypothesis based on $D_T$ with strong generalization to $D_V$:

$$\lambda^* \in \underset{\lambda \in \Lambda}{\arg\min} \, f(\lambda) := \underset{\lambda \in \Lambda}{\arg\min} \, \mathbb{E}_{(D_T, D_V) \sim D} \left[ \frac{1}{|D_V|} \sum_{(x,y) \in D_V} \ell\left(y, h_{\lambda, D_T}(x)\right) \right] . \quad (1)$$

While the HPO problem can be tackled naïvely via grid search or random search (Bergstra & Bengio, 2012), more sophisticated techniques, such as Bayesian optimization, are recommended today due to their higher efficiency and effectiveness (Turner et al., 2021; Bischl et al., 2023).

**Bayesian Optimization (BO)** (Močkus, 1975; Jones et al., 1998; Shahriari et al., 2016) is a model-based sequential optimization technique widely-used for sample-efficient HPO (Snoek et al., 2012; Falkner et al., 2018; Cowen-Rivers et al., 2022; Makarova et al., 2022; Bischl et al., 2023). BO is particularly well-suited for optimizing black-box functions $f$, which are costly to evaluate and have neither a closed-form solution nor any gradient information available.

The BO process begins with an initial design that aims to cover the hyperparameter configuration space diversely. It then proceeds, alternating between two main steps: (i) fitting a probabilistic surrogate model $\hat{f}$ to the set of observed evaluations, and (ii) selecting the next configuration

$$\lambda^{t+1} \in \underset{\lambda \in \Lambda}{\arg\max} \, \alpha_{\hat{f}}(\lambda) \quad (2)$$

where the acquisition function $\alpha_{\hat{f}} : \Lambda \to \mathbb{R}$ quantifies the utility of candidate points in balancing exploration and exploitation. In practice, and in line with standard GP-based BO (Rasmussen & Williams, 2006), the surrogate is typically trained on normalized observations (e.g., scaled to the $[0, 1]$ interval). This normalization stabilizes GP hyperparameter estimation and ensures that commonly used acquisition functions yield non-negative values by construction. A prominent example is Expected Improvement (EI) (Jones et al., 1998), which selects points expected to yield improvements over the best previously observed configuration, also called the incumbent. Formally, let $\hat{\lambda}$ denote the current incumbent of the black-box function, then EI at a hyperparameter configuration $\lambda$ is defined as

$$\alpha_{\hat{f}}^{EI}(\lambda) := \mathbb{E}\left[\max\left(f(\hat{\lambda}) - \hat{f}(\lambda), 0\right)\right],$$

where $\hat{f}(\lambda)$ is treated as a random variable representing the probability distribution at $\lambda$ as modeled by $\hat{f}$. In HPO, the black-box function $f$ typically corresponds to the empirical generalization error, estimating the expected loss as in Eq. 1, for example, via hold-out validation or cross-validation.

## 3 RELATED WORK

Approaches related to our work can be broadly grouped into (1) data-driven priors, (2) explicit user generated priors, and (3) interactive HPO frameworks.

**Data-Driven Priors** A substantial body of research leverages prior experience to configure the HPO process. This includes transfer learning across tasks (Swersky et al., 2013; Feurer et al., 2015; van Rijn & Hutter, 2018; Feurer et al., 2022), configuration space design (Perrone et al., 2019) and surrogate model configuration (Feurer et al., 2018). While effective, such approaches do not enable direct user interaction. Instead, they require users to trust automated knowledge transfer.

**Learning from User Priors** Another line of research explicitly incorporates user-specified beliefs. Bergstra et al. (2011) introduce fixed priors over the configuration space, whereas Souza et al. (2021b) estimate posterior-driven models, though both approaches are limited by their acquisition function compatibility. Ramachandran et al. (2020) warp the configuration space to emphasize promising regions, but this requires invertible priors and struggles with misleading ones. More recently, Hvarfner et al. (2022) propose $\pi$BO, which augments acquisition functions with priors in a flexible and robust way, whereas Mallik et al. (2023) extend HyperBand (Li et al., 2017) to balance

random sampling with user-defined priors. However, all of these methods restrict user input to the initialization phase and offer neither continuous control nor oversight over the optimization process.

**Interactive Hyperparameter Optimization**  Moving beyond priors provided at the beginning of the optimization process, some recent works integrate users more directly into the optimization process. Xu et al. (2024) allow users to reject candidate evaluations, while Adachi et al. (2024) let users choose among proposed alternatives. Seng et al. (2025) propose an alternative to the common BO framework that is based on probabilistic circuits and does not rely on an acquisition function. They sample the use of the prior in each iteration based on a binomial distribution decaying over time. Priors can be given at any time, but in contrast to our approach, they override previous information entirely, and the approach suffers from extrapolation issues. Complementary to these, Chang et al. (2025) introduce LLINBO working with LLM-generated candidate configurations to augment BO, with rejection schemes for ensuring robustness. Likewise, cooperative design optimization (Niwa et al., 2025) explores natural language interfaces where LLMs propose configurations, optionally guided by user input. Their additional user studies demonstrate that such interaction mitigates over-tuning to local optima while maintaining user agency.

In contrast, our proposed method `DynaBO` enables users to provide priors at any time during the HPO process, while remaining model-agnostic and acquisition function-compatible. Unlike previous methods, `DynaBO` not only integrates repeated user input but also detects misleading priors and can safeguard against those, thus advancing toward a more collaborative paradigm for HPO.

## 4 DYNAMIC PRIORS IN BAYESIAN OPTIMIZATION

In this section, we present how dynamic priors can be leveraged in Bayesian optimization (BO), providing the first component of our proposed method, `DynaBO`. To this end, we first generalize prior-weighted acquisition functions from a single prior to stacking multiple priors (Section 4.1). Afterward, we devise a method to detect misleading priors and safeguard against them (Section 4.2).

### 4.1 PRIOR-WEIGHTED ACQUISITION FUNCTION

Following Hvarfner et al. (2022), we integrate user-provided prior information on the location of the optimum by weighting the original acquisition function with a prior distribution. Thereby, we enable the optimization process to incorporate external knowledge dynamically, which may be provided by the user or taken from some other source of knowledge.

Given an acquisition function $\alpha$, and a user-specified prior $\pi : \Lambda \rightarrow (0, 1]$, the next point to be evaluated with respect to $f$ (e.g, as defined in Eq. 1) at time $t$ is selected as follows:

$$\alpha_{\hat{f}}^{\pi\text{BO}}(\lambda) := \alpha_{\hat{f}}(\lambda) \cdot \pi(\lambda)^{\beta/t},$$

where $\beta \in \mathbb{R}^+$ is a scaling hyperparameter. For $t \rightarrow \infty$, the weight induced by $\pi$ converges to 1 independent of the configuration $\lambda$ and $\beta$, that is, the effect of the prior diminishes over time.

In contrast to the work of Hvarfner et al. (2022), we suppose a finite sequence of user-specified priors $\{\pi^{(m)}\}_{m=1}^{M}$ provided at times $\{t^{(m)}\}_{m=1}^{M}$, $t^{(1)} < \ldots < t^{(M)} \leq T$. We define a dynamically-adapted acquisition function $\alpha_{\text{dyna}} : \Lambda \rightarrow \mathbb{R}$ by multiplying the sum of the priors:

$$\alpha_{\hat{f}}^{\text{dyna}}(\lambda) := \alpha_{\hat{f}}(\lambda) \cdot \sum_{m=1}^{M} \pi^{(m)}(\lambda)^{\beta/(t-t^{(m)})}$$

Figure 2: Acquisition function impact of priors $\pi^1, \pi^2, \pi^3$, provided at $t = 10, 20$, and 30, with $\pi(\lambda) = 0.5$.

By stacking priors, we can incorporate the information given at different time steps, and the priors are faded individually based on their age; that is, older priors are considered less important, as shown in Figure 2. The flexibility in incorporating multiple priors sets `DynaBO` apart from $\pi$BO (Hvarfner et al., 2022) and Priorband (Mallik et al., 2023), which consider a single initial prior, as well as Seng et al. (2025)'s approach, which considers one prior at a time.

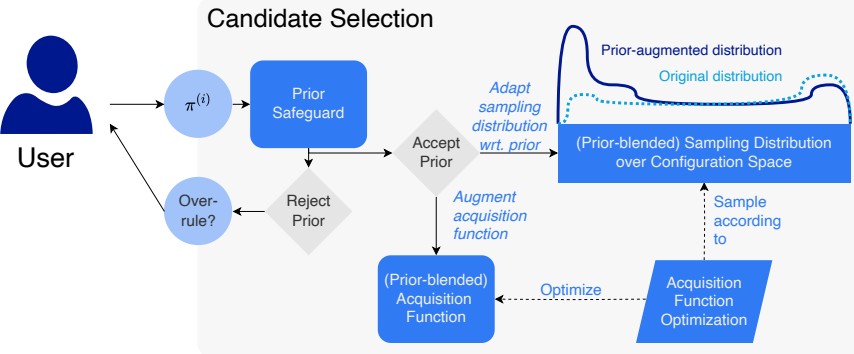

Figure 3: Illustration of the candidate selection process in `DynaBO`, incorporating a user-provided prior $\pi^{(i)}$. A safeguard mechanism evaluates the prior, determining whether to accept or reject it. If accepted, the candidate selection is biased by the prior; otherwise, the user can overrule the rejection.

**Acquisition Function Optimization**  If user priors are peaked in a small area of the configuration space, the resulting acquisition function $\alpha_{\mathrm{dyna}}$ could be harder to optimize, see Equation (2). To ensure that the acquisition function optimizer covers both user-suggested regions and generally unexplored areas of the configuration space adequately, we adopt a modified version of Hutter et al. (2011)'s combined local and random search. To ensure that a sufficient number of candidates are sampled close to the peaked prior, we adapt the sampling of the starting points of the local search to the prior distributions. To this end, each prior $\pi^{(m)}$ is assigned a weight $\omega_m = e^{\phi(t - t^{(m)})}$, which also decays with time. Then, a fraction $\omega_m \cdot \min\{\sum_{i=1}^{m} w_i, 0.9\}$ of candidate configurations are sampled according to prior $\pi^{(m)}$. For more details, we refer to Appendix B.2.

## 4.2 REJECTING PRIORS

To safeguard against misleading priors, potentially slowing down `DynaBO`, we propose a mechanism to reject priors based on their feasibility to yield improvements given the surrogate model $\hat{f}$ and an acquisition function $\xi$, which can but need not be the same as $\alpha$. For an overview of how the safeguard is embedded in `DynaBO`'s candidate selection routine, please refer to Figure 3. Specifically, we assess the promisingness of a prior $\pi^{(m)}$ by comparing the potential of the suggested region against the region around the best configuration found previously, that is, the current incumbent $\hat{\lambda}$.

Sampling a number of candidate configurations from normal distributions centered around the incumbent and the prior (denoted by $\mathcal{N}_{\hat{\lambda}}$ and $\mathcal{N}_{\pi^{(m)}}$, respectively), we compare the expected acquisition function values of both samples based on the surrogate model $\hat{f}$. We accept the prior if and only if the quality of the prior exceeds the quality of the incumbent area by a given threshold $\tau$:

$$\mathbb{E}_{\lambda \sim \mathcal{N}_{\pi^{(m)}}} \left[ \xi_{\hat{f}}(\lambda) \right] - \mathbb{E}_{\lambda \sim \mathcal{N}_{\hat{\lambda}}} \left[ \xi_{\hat{f}}(\lambda) \right] \geq \tau \ . \tag{3}$$

The higher $\tau$ is set, the fewer priors are accepted by `DynaBO`, but also the more misleading priors can be removed. Setting $\tau$ to a lower value lets `DynaBO` embrace user-provided priors more often, while making it more prone to misleading priors. For more details regarding our choice of $\tau$ and the rejection of priors, refer to Appendix B.3.

In practice, the acquisition function $\xi$ lives in the space of loss values, e.g., Lower Confidence Bound (LCB) or EI (Snoek et al., 2012). Since we want to reject priros based on their potential, we recommend utilizing LCB and setting $\tau$ with respect to the user's beliefs on the remaining optimization potential $f(\hat{\lambda}) - f(\lambda^*)$. In summary, this approach ensures that optimization resources are not wasted on poor suggestions, that is, regions which are already known to perform inferior to the incumbent region. We also expect that the prior rejection scheme would be used to warn against priors, but users would have the chance to overrule their rejection. For a sensitivity analysis on $\tau$, refer to Section 6.4.

## 5 THEORETICAL ANALYSIS

In the theoretical analysis of `DynaBO`, we establish convergence and robustness properties under multiple dynamically provided user priors, and quantify the conditions under which informative priors yield accelerated convergence. Extending the convergence results of $\pi$BO (Hvarfner et al., 2022), we demonstrate that our approach retains the almost sure convergence behavior of BO even when given misleading priors. At the same time, our method is capable of leveraging informative priors effectively to accelerate convergence. Note that we assume a *finite* prior set, and the utilization of UCB as an acquisition function. All proofs are provided in Appendix A. We follow standard convergence results for BO (Srinivas et al., 2012) augmented by the dynamic prior influence.

**Almost Sure Convergence of `DynaBO`**   We analyze the asymptotic behavior of `DynaBO` given assumptions on objective function regularity, finiteness and vanishing influence of priors, introduced formally in Appendix A, allowing priors to be selected dynamically and to vary in quality over time.

**Theorem 1** (Almost Sure Convergence of `DynaBO`). *Under the assumptions in Appendix A, the sequence of query points selected by `DynaBO`, $\{\lambda_t\}_{t=1}^{\infty} \subset \Lambda$, satisfies almost sure convergence to the global optimum; that is, irrespective of the variation in priors, the method converges to an optimal configuration $\lambda^*$ with probability one:*

$$\mathbb{P}\left( \lim_{t \to \infty} f(\lambda_t) = f(\lambda^*) \right) = 1.$$

**Robustness to Misleading Priors**   Asymptotically, the algorithm does not suffer degradation in performance even when faced with misleading priors. Formally, it is guaranteed that the best objective value found after $t$ iterations is, in the limit, no worse than that of vanilla BO, providing a safeguard against user-supplied noise or misguidance. This is described in the following theorem:

**Theorem 2** (Robustness to Misleading Priors). *Let $f_t^*$ be the optimal value of $f$ found after $t$ iterations using `DynaBO`, and let $f_{t,BO}^*$ denote the corresponding value for standard BO. Then:*

$$\limsup_{t \to \infty}(f_{t,BO}^* - f_t^*) \le 0.$$

**Acceleration of Convergence with Informative Priors**   While being robust to misleading priors, `DynaBO` is also designed to benefit from informative ones. When user-provided priors place high probability mass in regions near or containing the global optimum, the optimization process can concentrate evaluations more effectively within these promising areas. This focused search results in an improved convergence rate, captured by a regret bound dependent on the prior's concentration. The following result formalizes this intuition by showing that the regret bound depends only on the information gain within the neighborhood of the optimum:

**Theorem 3** (Acceleration of Convergence with Informative Priors). *Suppose there exists a prior $\pi^{(m)}$ such that the prior mass in a neighborhood $U_\epsilon(\lambda^*)$ of the global optimum $\lambda^*$ satisfies*

$$\int_{U_\epsilon(\lambda^*)} \pi^{(m)}(\lambda) \, d\lambda \ge 1 - \delta,$$

*for some small $\delta > 0$. Then, the expected cumulative regret $\mathbb{E}[R_T]$ of our method after $T$ iterations satisfies*

$$\mathbb{E}[R_T] = \mathcal{O}\left( \sqrt{T \beta_{UCB_T} \, \gamma_T(U_\epsilon)} \right),$$

*with high probability, where $\beta_{UCB_T}$ is a confidence parameter of the UCB algorithm. $\gamma_T(U_\epsilon)$ is the maximum information gain restricted to the neighborhood $U_\epsilon(\lambda^*)$, and $\gamma_T(U_\epsilon) < \gamma_T(\Lambda)$.*

## 6 EMPIRICAL EVALUATION

In this section, we present the empirical evaluation of `DynaBO`, analyzing its anytime performance across various black-box benchmark scenarios and for different qualities of priors. Section 6.1 describes how priors are constructed. The experiment setup is detailed in Section 6.2 and the results are presented in Section 6.3. To investigate `DynaBO`'s sensitivity with respect to the prior rejection threshold $\tau$, we conduct a sensitivity analysis in Section 6.4.

## 6.1 PRIOR CONSTRUCTION: EXPERT, INFORMATIVE, LOCAL, ADVERSARIAL

Inspired by the evaluation protocols of Souza et al. (2021a); Hvarfner et al. (2022); Mallik et al. (2023), and Seng et al. (2025), we construct artificial, data-driven priors. To ground our investigation in an analysis of well and poorly performing areas of the configuration space, we conduct an extensive search for every benchmark scenario and cluster the found configuration-loss pairs $(\lambda, \ell_\lambda) := (\lambda, f(\lambda))$ hierarchically via Gower's distance (Gower, 1971) into $n$ clusters. For each of the $n$ clusters, $c_1, \ldots, c_n$, we compute its centroid $\overline{c_i}$ and the median loss $\overline{\ell_{c_i}}$ of configurations contained. In the following, we assume the clusters to be ordered according to their median loss, that is, $\overline{\ell_{c_i}} \leq \overline{\ell_{c_j}}$ for $i < j$. For more information on the construction of clusters, we refer to Appendix B.

To obtain dynamic priors, considering the current incumbent $\hat{\lambda}$ and its loss $\ell_{\hat{\lambda}}$, we select the cluster $c^+$ and configuration $\lambda^+$ according to the following four prior policies simulating different aspects and levels of informativeness. The chosen configuration $\lambda^+$ is then used as the center of a normal distribution over the configuration space to guide the optimization process toward its broader region.

**Expert Priors** bias toward clusters spanning significantly-better regions of the configuration space, that is, $\ell_{\hat{\lambda}} \geq \overline{\ell_{c_i}}$. A cluster $c^+$ is sampled with probability $\mathbb{P}(c_i) \propto e^{0.1 \cdot i}$. From this cluster, we choose the best configuration $\lambda^+ \in \arg\min_{\lambda \in c^+} \ell_\lambda$ as the center of the normal distribution of our prior.

**Advanced Priors** bias toward clusters spanning better-performing regions of the configuration space, that is, $\ell_{\hat{\lambda}} \geq \overline{\ell_{c_i}}$. A cluster $c^+$ is sampled with probability $\mathbb{P}(c_i) \propto e^{0.15 \cdot i}$. From this cluster, we sample $\lambda^+ \in c^+$ randomly as the center of the normal distribution of our prior.

**Local Priors** bias toward well-performing clusters close to the current incumbent. To this end, the incumbents' Gower's distance (Gower, 1971) to each cluster $D^{gower}(\hat{\lambda}, \overline{c_i})$ is utilized to select the 10 closest, later considered clusters $C^+$. The cluster with the lowest median loss $c^+ \in \arg\min_{c \in c^+} \overline{\ell_c}$ is selected, and the prior center $\lambda^+ \in c^+$ is sampled randomly. In contrast to previous evaluations of prior-guided BO (Souza et al., 2021a; Hvarfner et al., 2022; Mallik et al., 2023), we hypothesize that such local priors, exploiting knowledge about the location of current incumbents, are more similar to human behavior. Simultaneously, local priors are closer to already observed data used for fitting the surrogate model, facilitating informed acquisition.

**Adversarial Priors** bias toward sampling configurations in poorly performing regions of the configuration space. For that, $c^+$ is randomly sampled from the five clusters with the worst median loss. The center of the prior is set to $\lambda^+ \in \arg\max_{\lambda \in c^+} \ell_\lambda$. This is meant to simulate the worst case in which a human user provides priors based on wrong assumptions.

## 6.2 EXPERIMENT SETUP

Although our theoretical analysis focuses on LCB, we conduct our experimental evaluation using the common EI acquisition function. As baselines, we consider the state-of-the-art approach $\pi$BO and vanilla Bayesian optimization (vanilla BO), as implemented in the SMAC3 library (Lindauer et al., 2022). $\pi$BO allows the user to provide prior information before the optimization process, whereas vanilla BO is without any user guidance. We chose to disregard BoPro (Souza et al., 2021b) as a baseline, since its substantially dominated by $\pi$BO. We evaluated the approaches both with Gaussian processes (GPs) and random forests (RFs). In accordance with the results of Eggensperger et al. (2021), SMAC with RFs showed stronger results overall and can handle mixed spaces natively; the results with GPs are provided in Appendix D.1. We utilize LCB as an acquisition function for prior rejection. Details on the competitors are provided in Appendix C.2.

In our experiments, we evaluate various models, ranging from traditional to complex deep learning models, discussed in Appendix C.1. For each model and dataset, we repeat every HPO run with 30 different seeds. Given the same seed, `DynaBO` and $\pi$BO will sample identical (first) priors. For simple models, we allow for 200 trials and four priors after 50, 90, 130, and 170 trials. Considering the comparatively large run-time of complex models, we reduce their trial budget to 50 with priors after iterations 13, 23, 33, and 43. Experiments for random prior times are presented in Appendix D.3. Following the recommendations by Hvarfner et al. (2022), $\beta$ is initialized as $N/10$ with $N$ denoting the number of overall trials. All experiments discussed in this paper were executed on HPC nodes

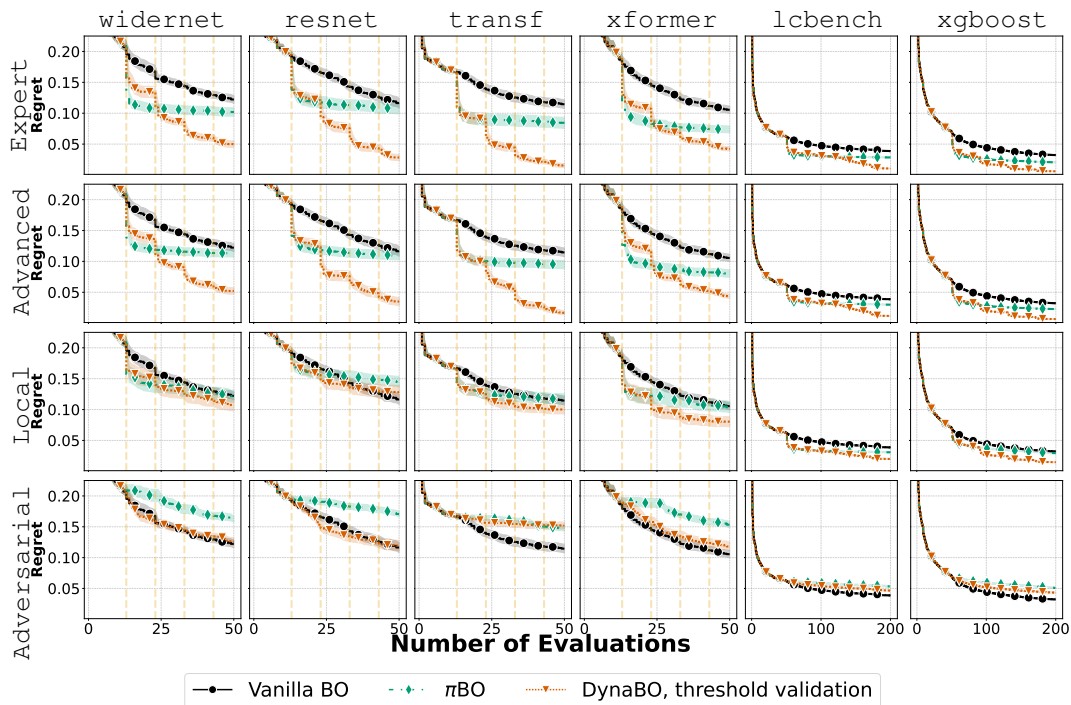

Figure 4: Mean regret for `lcbench`, `xgboost`, and `PD1` using `Expert`, `Advanced`, `Local`, and `Adversarial` priors. Priors are provided at vertical lines. The shaded areas visualize the standard error. For `lcbench` and `xgboost`, the plots average all datasets. The results indicate `DynaBO` outperforming $\pi$BO and remaining competitive to vanilla BO for adversarial priors.

equipped with 2 Intel(R) Xeon(R) Platinum 8470 @2.0GHz processors and 488GiB RAM, of which 2 CPU cores and 6GB RAM were allocated per run.

## 6.3 RESULTS

In Figure 4, we present anytime performance plots where the mean regret of the best found incumbent, wrt. preceding exploration detailed in Appendix B, is plotted over the number of evaluations of the black box function $f$.

**Expert and Advanced Priors**  Generally speaking, `DynaBO` outperforms vanilla BO, and $\pi$BO significantly in both anytime and final performance. On `widernet` and `xformer`, `DynaBO` is predominated by $\pi$BO until the second prior is provided, due to overly-cautious rejection of priors. Generally, expert priors provide a larger performance boost than advanced priors.

**Local Priors**  For local priors, we see similar results, but with a reduced gain through the provided priors. In the case of `resnet`, local priors reduce rather than improve performance for both $\pi$BO and `DynaBO`. However, this impacts $\pi$BO more than `DynaBO`.

**Adversarial Priors**  Generally speaking, adversarial priors result in performance degradation for $\pi$BO, as well as `DynaBO`. However, our rejection mechanism results in a significant performance boost except for `transformer_1m1b`. The recovery from adversarial priors is analyzed in Figure 5 using an increased budget. Here `DynaBO` outperforms $\pi$BO if equipped with Gaussian processes, and `DynaBO`'s prior rejection scheme is necessary if random forests are used as surrogate model. Additional plots, considering each scenario individually, are provided in Appendix D.4.

As a general trend, `DynaBO` performs equal or superior to $\pi$BO. Additionally, `DynaBO`'s rejection mechanism boosts performance for both Local and Adversarial priors, while only marginally reducing solution quality for Expert and Advanced priors. Other results using GPs for the `PD1` benchmarks (Wang et al., 2024), presented in Appendix D.1, further support the analysis above. In

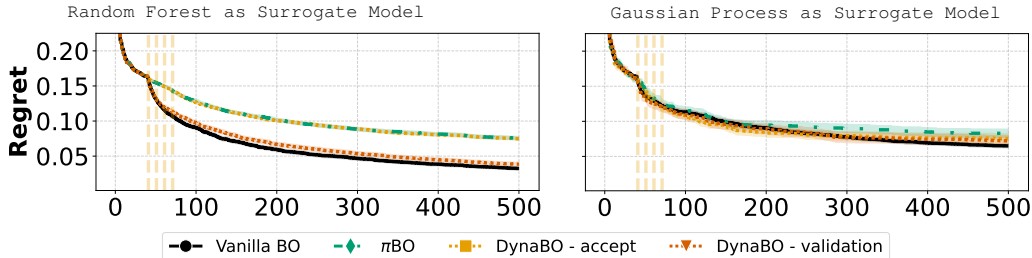

Figure 5: Anytime regret for `PD1` averaged over 30 seeds, and scenarios comparing `vanilla BO`, $\pi$BO, `DynaBO-accept` all priors, and `DynaBO` with validation (`DynaBO-validation`).

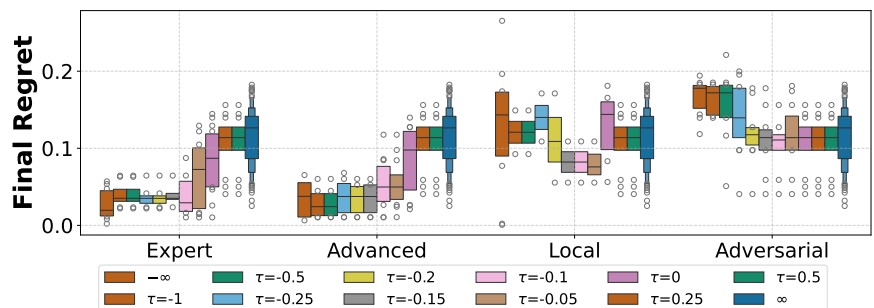

Figure 6: Sensitivity analysis of different thresholds $\tau$. Setting $\tau = -\infty$ accepts all, while setting $\tau = \infty$ rejects all priors. The boxplots contain the merged results from all scenarios.

Appendix D.7 we investigate the effect of decaying the prior faster, or slower. The results show that a linear prior decay is an adequate choice.

### 6.4 Sensitivity Analysis of the Prior Rejection Criterion

Our prior rejection scheme utilizes a threshold $\tau$, encoding the minimum estimated average improvement over the current incumbent needed to accept a prior. This improvement is quantified with LCB, one option for Equation (3). When $\tau < 0$, priors are accepted, even if they are predicted to be misleading; $\tau > 0$ ensures that only priors of ample potential are accepted.

To study the impact of $\tau$ on `DynaBO`, we conduct a sensitivity analysis on PD1's (Wang et al., 2024) optimization scenarios, shown in Figure 6 (further results in Appendix D). As anticipated, $\tau$ enables a tradeoff between being permissive to potentially helpful and rejecting misleading priors. While $\tau$ could in principle be customized to reflect the user's confidence or expertise level, we find that setting $-0.25 \leq \tau \leq -0.05$ strikes a good balance. The preceding experiments utilize $\tau = -0.15$.

### 7 Conclusion

In this work, we presented `DynaBO`, a method that seamlessly incorporates user beliefs into Bayesian optimization (BO) for hyperparameter optimization. To this end, `DynaBO` can adapt any acquisition function by incorporating priors at any point in the process. To ensure robustness, `DynaBO` includes a safeguard that rejects misleading priors. Empirical results demonstrate that `DynaBO` outperforms both vanilla BO and $\pi$BO with a static prior, while also offering theoretical guarantees of robustness and effectiveness—even without the safeguard. Nonetheless, the safeguard strengthens empirical resilience against misleading user guidance.

However, while our empirical evaluation considers different kinds of user priors, they are generated synthetically. Future work should focus on expanding the types of priors supported, carrying out user studies, and developing improved rejection mechanisms, for example, by evaluating prior-based configurations. Additionally, analyzing the effect of different surrogate models on prior handling in detail and extending `DynaBO` to multi-fidelity optimization similar to Mallik et al. (2023) are

worth exploring. Lastly, we aim to enhance `DynaBO` with explainable AI (XAI) and large language models (LLMs) to support more transparent, user-friendly, low/no-code interfaces, fostering more intuitive collaboration between users and HPO approaches. Lastly, as in related papers (Hvarfner et al., 2022; Seng et al., 2025), `DynaBO` focuses on black-box optimization and should be extended to multi-fidelity optimization.

**Reproducibility Statment**    We have taken several steps to ensure the reproducibility of our results. We follow standard methodological practices, including the use of best practices for random seeding. An additional, in-depth description of the experimental setup is provided in Appendix C. A detailed description of the conducted data generation runs is provided in Appendix B. Furthermore, we provide a reference implementation via GitHub: `https://anonymous.4open.science/r/DynaBO-EBB3/`.

**Ethical Considerations.**    We believe that our work does not raise any specific ethical concerns beyond the standard considerations associated with research in machine learning.

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

# ORGANIZATION OF TECHNICAL APPENDICES AND SUPPLEMENTARY MATERIAL

The appendix is split into four main parts. In Appendix A, we prove our theoretical guarantees. In Appendix B, we provide a detailed discussion of our prior construction and selection scheme. In Appendix C, we discuss the experimental setup. Lastly, in Appendix D, we provide additional evaluation results.

## A    THEORETICAL GUARANTEES

For all the subsequent proofs, we make the following assumptions:

**A1 - Objective Function Regularity:** The objective function $f_{\text{obj}} : \Lambda \rightarrow \mathbb{R}$ is bounded and Lipschitz-continuous on a compact domain. Additionally, $f_{\text{obj}}$ is assumed to be a realization from a Gaussian process prior with mean $m : \Lambda \rightarrow \mathbb{R}$ and positive definite kernel $k(\lambda, \lambda')$.

W.l.o.g., we minimize the objective function. However, for the theoretical analysis, we consider the maximization of the equivalent utility function $f(\lambda) := C - f_{\text{obj}}(\lambda)$, where the constant $C$ is chosen to ensure $f(\lambda) > 0$ for all $\lambda \in \Lambda$. This affine transformation preserves the global optimum and guarantees the strict positivity required for multiplicative re-weighting via the sum of priors.

**A2 - Finiteness of User Priors:** Let $\{(t^{(m)}, \pi^{(m)})\}_{m=1}^{M}$ be a finite series of user-specified prior functions $\pi^{(m)} : \Lambda \rightarrow (0, 1]$ with iteration indices $t^{(m)} \in \mathbb{N}$ such that $t^{(1)} < ... < t^{(M)}$.

**A3 - Vanishing Influence of Priors:** There exists $\beta \in \mathbb{R}$ such that for all indices $m$ and $\lambda \in \Lambda$

$$\lim_{t \to \infty} \pi^{(m)}(\lambda)^{\beta/(t-t^{(m)})} = 1.$$

That is, the multiplicative influence of the prior function on the acquisition criterion diminishes to unity with increasing number of iterations.

Additionally, we assume Upper Confidence Bound (UCB) as an acquisition function for maximization of the utility function.

### A.1    THEOREM - ALMOST SURE CONVERGENCE OF DYNABO

Given Assumptions A1, A2 and A3, the DynaBO algorithm converges almost surely to the global optimum of $f$; meaning, the sequence of points $(\lambda_t)_{t \geq 1}$ selected by DynaBO satisfies

$$f(\lambda_t) \xrightarrow{\text{a.s.}} f(\lambda^*).$$

*Proof.* The proof follows standard convergence results for Bayesian optimization (BO) (Srinivas et al., 2012) augmented by the dynamic influence of priors.

Let $\lambda^* \in \arg\max_{\lambda \in \Lambda} f(\lambda)$ be a global maximizer of $f$. At iteration $t > t^{(M)}$, the DynaBO acquisition function is defined as

$$\alpha_{\text{dyna}}(\lambda, t) := \alpha(\lambda, t) \sum_{m=1}^{M} \pi^{(m)}(\lambda)^{\beta/(t-t^{(m)})},$$

where $\alpha(\lambda, t)$ denotes a standard GP-UCB-acquisition function.

To prove convergence, we must show that the instantaneous regret $r_t := f(\lambda^*) - f(\lambda_t)$ converges to zero almost surely. We prove this by analyzing the event $\mathcal{E}_t = \{\lambda_t \in D_\epsilon\}$ at any time $t$, where $D_\epsilon = \{\lambda \in \Lambda \mid f(\lambda) \leq f(\lambda^*) - \epsilon\}$ is a sub-optimal region. First, the UCB parameter $\beta_{\text{UCB}_t}$ is chosen such that the probability of the confidence bounds failing, $\mathbb{P}(\mathcal{F}_t)$, is summable. By the Borel-Cantelli lemma, this occurs only finitely many times, almost surely.

Second, we analyze the event $\mathcal{E}_t$ on the high-probability event $\mathcal{F}_t^C$, complement of $\mathcal{F}_t$, where the bounds hold. If the algorithm selects a suboptimal sample $\lambda_t \in D_\epsilon$, it must be that $\alpha_{\text{dyna}}(\lambda_t, t-1) \geq \alpha_{\text{dyna}}(\lambda^*, t-1)$. Let $P(\lambda, t) := \sum_{m=1}^{M} \pi^{(m)}(\lambda)^{\beta/(t-t^{(m)})}$. By definition:

$$\alpha(\lambda_t, t-1) \cdot P(\lambda_t, t-1) \geq \alpha(\lambda^*, t-1) \cdot P(\lambda^*, t-1).$$

Since the bounds hold, $\alpha(\lambda, t-1)$ satisfies $\alpha(\lambda_t, t-1) \leq f(\lambda_t) + 2\beta_t^{1/2}\sigma_{t-1}(\lambda_t)$ and $\alpha(\lambda^*, t-1) \geq f(\lambda^*)$. Substituting these gives:

$$(f(\lambda_t) + 2\beta_t^{1/2}\sigma_{t-1}(\lambda_t)) \cdot P(\lambda_t, t-1) \geq f(\lambda^*) \cdot P(\lambda^*, t-1).$$

By Assumption A3 and the finiteness of $M$, $\lim_{t\to\infty} P(\lambda, t) = 1$. This convergence is uniform on the compact set $\Lambda$, by Dini's Theorem (Jost, 2005), given continuity of $\pi^{(m)}$ per the prior construction as a normal distribution. Therefore, for any $\delta > 0$, there exists a $T$ such that for all $t > T$, $P(\lambda, t) \in (1 - \delta, 1 + \delta)$ for all $\lambda \in \Lambda$. For $t > T$, we can bound the terms $P(\lambda_t, t-1) \le (1+\delta)$ and $P(\lambda^*, t-1) \ge (1-\delta)$ :

$$(f(\lambda_t) + 2\beta_t^{1/2}\sigma_{t-1}(\lambda_t))(1+\delta) \ge f(\lambda^*)(1-\delta).$$

As $f(\lambda_t) \le f(\lambda^*) - \epsilon$ (since $\lambda_t \in D_\epsilon$), we have:

$$(f(\lambda^*) - \epsilon + 2\beta_t^{1/2}\sigma_{t-1}(\lambda_t))(1+\delta) \ge f(\lambda^*)(1-\delta).$$

Rearranging for the variance term (and assuming $f, \epsilon > 0$ for simplicity):

$$2\beta_t^{1/2}\sigma_{t-1}(\lambda_t)(1+\delta) \ge f(\lambda^*)(1-\delta) - (f(\lambda^*) - \epsilon)(1+\delta)$$

$$\Leftrightarrow 2\beta_t^{1/2}\sigma_{t-1}(\lambda_t)(1+\delta) \ge \epsilon(1+\delta) - 2\delta f(\lambda^*).$$

As this must hold for any $\delta > 0$ (by picking $t$ large enough), we can let $\delta \to 0$, which implies that for $t$ sufficiently large, the condition for sampling in $D_\epsilon$ requires:

$$2\beta_t^{1/2}\sigma_{t-1}(\lambda_t) \ge \epsilon.$$

We now show that this condition can only be met a finite number of times. Let $t_n$ be the time of the $n$-th sample drawn from the sub-optimal region $D_\epsilon$. The UCB parameter $\beta_{t_n}$ grows as $O(\log t_n)$ (Srinivas et al., 2012). Crucially, Assumption A1 guarantees that the posterior variance at that point, $\sigma_{t_n-1}(\lambda_{t_n})$, converges to 0 as $n \to \infty$ (Srinivas et al., 2012). Because the variance $\sigma_{n-1}$ converges to 0 faster than the $\beta_{t_n}^{1/2}$ term grows (Srinivas et al., 2012), their product must also converge to zero:

$$\lim_{n\to\infty} 2\beta_{t_n}^{1/2}\sigma_{t_n-1}(\lambda_{t_n}) = 0.$$

This directly implies that for any $\epsilon > 0$, the condition $2\beta_t^{1/2}\sigma_{t-1}(\lambda_t) \ge \epsilon$ must eventually be violated. In other words, there exists a finite sample count $N_\epsilon$ after which the necessary condition for sampling in $D_\epsilon$ is permanently false.

In conclusion, the bounds fail only finitely often almost surely, and any sub-optimal region $D_\epsilon$ is sampled only a finite number of times (a.s.). Therefore, the event $\mathcal{E}_t$ occurs only finitely many times, almost surely. Since this holds for any $\epsilon > 0$, it follows that $r_t \xrightarrow{\text{a.s.}} 0$ and

$$f(\lambda_t) \xrightarrow{\text{a.s.}} f(\lambda^*).$$

$\square$

## A.2  COROLLARY - ROBUSTNESS TO MISLEADING PRIORS

Under Assumptions A1–A3, the asymptotic performance of DynaBO is robust to any finite set of misleading or incorrect priors; that is, the presence of such priors does not degrade convergence compared to standard BO. Formally, let

$$f_t^* := \max_{1\le i\le t} f(\lambda_i) \quad \text{and} \quad f_{t,BO}^* := \max_{1\le i\le t} f(\lambda_i^{BO})$$

be the maximum function values found by DynaBO and standard BO, respectively, at time $t$. Then,

$$\limsup_{t\to\infty}(f_{t,BO}^* - f_t^*) \le 0.$$

*Proof.* By Assumption A3, analogously to Theorem A.1, the DynaBO acquisition function

$$\alpha_{\text{dyna}}(\lambda, t) = \alpha(\lambda, t) \sum_{m=1}^{M} \pi^{(m)}(\lambda)^{\beta/(t-t^{(m)})}$$

converges uniformly over $\Lambda$ to the standard BO acquisition function:

$$\sup_{\lambda \in \Lambda} |\alpha(\lambda, t) - \alpha_{\text{dyna}}(\lambda, t)| \to 0 \quad \text{as } t \to \infty.$$

Hence, the sequence of query points $\{\lambda_t\}$ selected by DynaBO converges in distribution to those of standard BO, $\{\lambda_t^{\text{BO}}\}$. By Assumption 1, the Lipschitz-continuity of $f$, the Continuous Mapping Theorem implies

$$f(\lambda_t) \xrightarrow{d} f(\lambda_t^{\text{BO}}).$$

Since $\{f_t^*\}_{t \geq 1}$ and $\{f_{t,BO}^*\}_{t \geq 1}$ are bounded non-decreasing sequences, their limits exist. Therefore,

$$\limsup_{t \to \infty}(f_{t,BO}^* - f_t^*) \leq 0,$$

establishing the asymptotic robustness of DynaBO to misleading priors. $\square$

### A.3 THEOREM - ACCELERATION OF CONVERGENCE WITH INFORMATIVE PRIORS

Under Assumptions A1-A3, suppose there exists a user prior $\pi^{(m)}$ such that the prior density $\pi^{(m)}(\lambda)$ places a substantial mass in a neighborhood $U_\epsilon(\lambda^*)$ containing the global optimum $\lambda^*$:

$$\int_{U_\epsilon(\lambda^*)} \pi^{(m)}(\lambda)d\lambda \geq 1 - \delta$$

for small $\delta > 0$. Then, the DynaBO algorithm achieves an improved upper bound on cumulative regret over $T$ iterations, specifically

$$R_T := \sum_{t=1}^{T}(f(\lambda^*) - f(\lambda_t)) = O\big(\sqrt{T\beta_{\text{UCB}_T}\gamma_T(U_\epsilon)}\big).$$

Here, $\gamma_T(U_\epsilon)$ denotes the maximum information gain restricted to the neighborhood $U_\epsilon(\lambda^*)$, satisfying $\gamma_T(U_\epsilon) < \gamma_T(\Lambda)$. This implies asymptotically faster convergence rates than standard BO without informative priors.

*Proof.* Define the index sets

$$\mathcal{I}_{U_\epsilon} := \{1 \leq t \leq T \mid \lambda_t \in U_\epsilon\} \quad \text{and} \quad \mathcal{I}_{\Lambda \setminus U_\epsilon} := \{1 \leq t \leq T \mid \lambda_t \notin U_\epsilon\}.$$

Similarly to Theorem A.1, BO regret bounds using the GP kernel and mutual information gain apply to $U_\epsilon$ (Srinivas et al., 2012):

$$\sum_{t \in I_{U_\epsilon}}(f(\lambda^*) - f(\lambda_t)) \leq C\sqrt{|I_{U_\epsilon}|\beta_{\text{UCB}_{|I_{U_\epsilon}|}}\gamma_{|I_{U_\epsilon}|}(U_\epsilon)} \leq C\sqrt{T\beta_{\text{UCB}_T}\gamma_T(U_\epsilon)},$$

where $C$ is kernel-dependent.

Now, let $B := \max_{\lambda \in \Lambda}(f(\lambda^*) - f(\lambda))$ be the maximum instantaneous regret. Then,

$$\sum_{t \in \mathcal{I}_{\Lambda \setminus U_\epsilon}}(f(\lambda^*) - f(\lambda_t)) \leq B|\mathcal{I}_{\Lambda \setminus U_\epsilon}|.$$

The region $\Lambda \setminus U_\epsilon$ is sub-optimal. It is a result of UCB analysis (P. Auer & Fischer, 2002) that the expected number of times a sub-optimal arm is sampled, $\mathbb{E}[|\mathcal{I}_{\Lambda \setminus U_\epsilon}|]$, is bounded logarithmically in time, i.e., $O(\log T)$. Since $O(\log T)$ grows strictly slower than $T$, it is $o(T)$. Therefore,

$$\mathbb{E}\left[\sum_{t \in \mathcal{I}_{\Lambda \setminus U_\epsilon}}(f(\lambda^*) - f(\lambda_t))\right] \leq B\mathbb{E}[|\mathcal{I}_{\Lambda \setminus U_\epsilon}|] = o(T).$$

By linearity of expectation,

$$\mathbb{E}[R_T] = \mathbb{E}\left[\sum_{t=1}^{T}(f(\lambda^*) - f(\lambda_t))\right] = \mathbb{E}\left[\sum_{t \in \mathcal{I}_{U_\epsilon}}(f(\lambda^*) - f(\lambda_t))\right] + \mathbb{E}\left[\sum_{t \in \mathcal{I}_{\Lambda \setminus U_\epsilon}}(f(\lambda^*) - f(\lambda_t))\right].$$

Plugging in the previous results yields

$$\mathbb{E}[R_T] \leq C\sqrt{T\beta_{\text{UCB}_T}\gamma_T(U_\epsilon)} + o(T).$$

As for the asymptotic implications: Since $\gamma_T(U_\epsilon) < \gamma_T(\Lambda)$, the cumulative regret bound improves over standard BO without informative priors when $\delta$ is sufficiently small. $\square$

# B    PRIOR HANDLING

## B.1    PRIOR CONSTRUCTION AND SELECTION

As mentioned in Section 2, given a learner (or learning algorithm) $A$, dataset $D = (D_T, D_V)$, and a loss function $\ell$, HPO aims to find a well-performing hyperparameter configuration $\hat{\lambda}$. A learning algorithm $A$, configured with $\hat{\lambda}$, and applied to $D$, results in a hypothesis $h_{\hat{\lambda}, D}$ minimizing the loss function $\ell$.

As discussed in Section 6.1, we construct priors based on data collected through Bayesian optimization runs. These data generation runs are conducted as follows:

1. Generate prior data: For each learner $A$, dataset $D$ combination, execute explorative Bayesian optimization runs with both the more greedy Expected Improvement (EI) and more explorative Lower Confidence Bounds (LCB) (Papenmeier et al., 2025). For each acquisition function, run 10 seeds for a budget of $5,000$ iterations. Then, for every algorithm, dataset combination, concatenate the lists of preliminary incumbents and assemble a joint list sorted by losses $\ell_\lambda$:

$$I_{A,D} = [(\lambda_1, \ell_{\lambda_1}), (\lambda_2, \ell_{\lambda_2}), \ldots, (\lambda_n, \ell_{\lambda_n})].$$

   Note the abuse of notation: We denote with $\ell_\lambda$ the loss on the validation set as introduced in Equation (1):

$$\ell_\lambda = \left[ \frac{1}{|D_V|} \sum_{(x,y) \in D_V} \ell\left(y, h_{\lambda, D_T}(x)\right) \right].$$

2. To ensure that also non-well-performing areas of the configuration space are covered, $I_{A,D}$ is supplemented with $n$ non-incumbent configurations and their loss.

3. Due to structured configuration spaces, some hyperparameters may not be active. In the case of our experiments, this only occurs for numeric hyperparameters. These values are filled with $-1$.

4. Create clusters of configurations of preliminary incumbents in the configuration space: For each learner $A$, dataset $D$ combination, cluster the incumbent configurations into 100 clusters

$$C_{A,D} = \{c_1, c_2, ..., c_{100}\} \quad \text{with} \quad c_i = \{(\lambda_{c_i^1}, \ell_{\lambda_{c_i^1}}), (\lambda_{c_i^2}, \ell_{\lambda_{c_i^2}}), ...\}$$

   using Agglomerative Clustering with Gower's Distance (Gower, 1971) and Ward Linkage. For each cluster, compute a centroid $\overline{c_i}$ and the median performance $\overline{\ell_{c_i}}$.

During optimization of $A$ on dataset $D$, priors are generated dynamically.

1. Sample prior configuraiton $\lambda^+$ as described in Section 6.1.

2. Build prior: We hypothesize that with each prior provided to `DynaBO`, the confidence of a user would grow. In our synthetic prior generation, we therefore build the $k$-th prior $\pi^k$ as follows: For each numerical hyperparameter $\lambda_j$ with lower bounds $\lambda_1^l, \lambda_2^l, ..., \lambda_d^l$ and upper bounds $\lambda_1^u, \lambda_2^u, ..., \lambda_d^u$, set

$$\pi^m = [\mu_j, \sigma_j]_{j=1}^d = \left[ \left( \lambda_j^+, \frac{|\lambda_j^u - \lambda_j^l|}{k \cdot 5} \right) \right]_{j=1}^d. \tag{4}$$

3. For each categorical hyperparameter $\lambda_j = \lambda_j^+$.

As mentioned in Section 6.2, we provide four priors for evaluations on YAHPO Gym, and four priors $\pi^1, \pi^2, \pi^3, \pi^4$ for evaluations on PD1, respectively.

## B.2    FACILITATING PRIOR BEHAVIOR

**Numerical Stability**    To ensure numerical stability and to maintain the impact of the initial acquisition function, we clip priors at 1e-12 and thereby ensure that priors take values in $(0, 1]$. Furthermore, due to the decaying mechanism, priors converge to 1 with increasing $t$.

**Adapting Candidate Sampling** Furthermore, to facilitate sampling candidate configurations with a high prior impact, we adapt SMAC3's (Lindauer et al., 2022) Local and Random Search candidate generation. If no prior is provided, the standard SMAC acquisition function samples 5000 random configurations, ranks them according to the acquisition function, and selects the 10 best-ranked configurations as starting points for local search among other candidates.

Suppose a finite sequence of user-specified priors $\{\pi^{(m)}\}_{m=1}^{M}$ is provided at times $\{t^{(m)}\}_{m=1}^{M}$, with $t^{(1)} < ... < t^{(M)} \leq T$, then the random configurations are replaced as follows:

- At iteration $t$ prior $\pi^{(m)}$ is associated with a weight $\omega_m = e^{-0.126 \cdot (t - t^{(m)})}$.

- If $\sum\limits_{m=1}^{M} \leq 0.9$, each prior is used to sample $\lfloor \omega_m + 0.5 \rfloor$ configurations. The rest are sampled uniformly at random.

- If $\sum\limits_{m=1}^{M} > 0.9$, each prior is used to sample $\left\lfloor \dfrac{\omega_w}{\sum_{j=1}^{M} \omega_j} + 0.5 \right\rfloor$ configurations. The rest are sampled uniformly at random.

### B.3 FURTHER DETAILS ON THE REJECTION CRITERION

To apply our prior rejection scheme

$$\mathbb{E}_{\lambda \sim \mathcal{N}_{\pi^{(m)}}} \left[ \alpha_{\hat{f}}(\lambda) \right] - \mathbb{E}_{\lambda^+ \sim \mathcal{N}_{\hat{\lambda}}} \left[ \alpha_{\hat{f}}(\lambda^+) \right] \geq \tau \ . \tag{5}$$

On categorical hyperparameters, we conduct the following adaptations: For the configurations sampled according to the prior $\lambda \sim \mathcal{N}_{\pi^{(m)}}$, categorical values are sampled according to the provided weights. For the configurations sampled in the area around the incumbent $\lambda^+ \sim \mathcal{N}_{\hat{\lambda}}$, the incumbents configuraiton is utilized.

# C   DETAILED EXPERIMENTAL SETUP

## C.1   SUMMARY OF THE BENCHMARK

Rather than training many models with different hyperparameter configurations, we use surrogate models provided by (Pfisterer et al., 2022) for XGBoost (`xgboost`) (Chen & Guestrin, 2016) and multi-layer perceptrons (`lcbench`) dubbed traditional machine learning. For complex architectures, we utilize surrogates trained for Mallik et al. (2023) based on data collected by Wang et al. (2024). We consider a wide ResNet (He et al., 2016) (`widernet`) trained on CIFAR10 (Krizhevsky, 2009), a ResNet (He et al., 2016) (`resnet`) trained on ImageNet (Deng et al., 2009), a transformer (Vaswani et al., 2017) (`transf`) trained on translatewmt (Bojar et al., 2015), and a transformer trained on WMT15 (`xformer`) (Bojar et al., 2015). A learner, searchspace, and dataset overview is provided in Table 1.

Table 1: An overview of the evaluated *scenarios*, each with the considered configuration *configuration space* type and number of *datasets*, with which each scenario was evaluated.

| Scenario | Configuration Space | # Datasets |
|---|---|---|
| rbv2_xgboost | 14D: Mixed | 119 |
| lcbench | 7D: Numeric | 34 |
| cifar100_wideresnet_2048 | 4D: Numeric | 1 |
| imagenet_resnet_512 | 4D: Numeric | 1 |
| lm1b_transformer_2048 | 4D: Numeric | 1 |
| translatewmt_xformer_64 | 4D: Numeric | 1 |

Our experiments are scheduled, and the results are logged in a MySQL database using the PyExperimenter library (Tornede et al., 2023).

## C.2   COMPETITOR SETUP

Our experiments are built on top of SMAC3 (Lindauer et al., 2024), for vanilla BO, $\pi$BO, and `DynaBO`. We utilize the Hyperparameter Optimization Facade, but deactivate its standard $\log$ transformations. We also refit the surrogate after every evaluated configuration.

## C.3   IMPLEMENTATION DETAILS OF PRIOR REJECTION

Following the principle of optimism in the face of uncertainty, we assess the potential of both regions in terms of their lower confidence bounds (LCBs) (Agrawal, 1995):

$$LCB(\lambda) = (-1) \cdot (\mu(\lambda) - \kappa\sigma(\lambda)),$$

where $\mu(\cdot)$ denotes the mean predicted with $\hat{f}$ for $\lambda$, and $\sigma(\cdot)$ the uncertainty in terms of standard deviation. This way, we interpret all uncertainty the surrogate model may have at a configuration $\lambda$ as the potential for its performance to achieve an improvement at all. Intuitively, the LCB criterion allows us to quantify the explorative potential of the prior region as opposed to the exploitative potential close to the current incumbent, since uncertainty close to the incumbent is typically low.

Provided a prior $\pi^{(m)}$ with mean $\mu^{(m)}$, and standard deviation $\sigma^{(m)}$ as in Equation (4), our initialization of the rejection criterion detailed in Equation (3)

$$\mathbb{E}_{\lambda \in \mathcal{N}_{\pi^{(m)}}}[\alpha(\lambda)] - \mathbb{E}_{\lambda \in \mathcal{N}_{\hat{\lambda}}}[\alpha(\lambda)] \geq \tau \qquad (6)$$

utilizes 500 configurations from normal distributions for both $\mathcal{N}_{\pi^{(m)}} \sim (\mu^{(m)}, \sigma^{(m)})$, and $\mathcal{N}_{\hat{\lambda}} \sim (\hat{\lambda}, \mu^{(m)})$. Our main experiments utilize $\tau = -0.15$.

# D  ADDITIONAL EMPIRICAL RESULTS

Our additional experimental results focus on further validating our approach. Appendix D.1 contains the results on the PD1 benchmark (Wang et al., 2024) with gaussian processes as a surrogate. For an easier comparison, the corresponding random forest results are provided in Appendix D.2. Appendix D.3 contains the results of experiments conducted with randomly sampled prior locations. All experiments shown here are conducted on the PD1 benchmark.

## D.1  GAUSSIAN PROCESS MAIN RESULTS

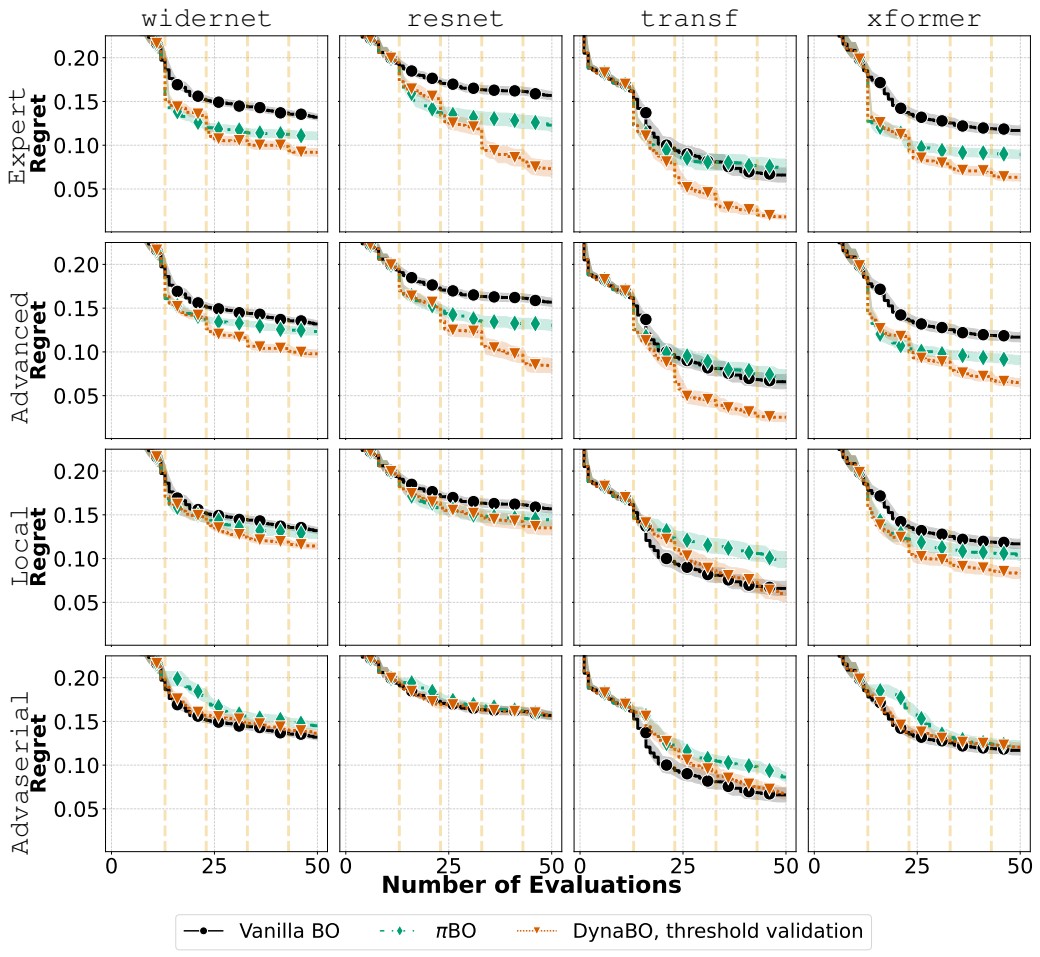

Figure 7: Mean regret for and `PD1` using Expert, Advanced, Local, and Adversarial priors, with Gaussian processes as surrogate models. Priors are provided at vertical lines. The shaded areas visualize the standard error. The results indicate `DynaBO` outperforming $\pi$BO and remaining competitive with vanilla BO for adversarial priors.

## D.2 RANDOM FOREST MAIN RESULTS

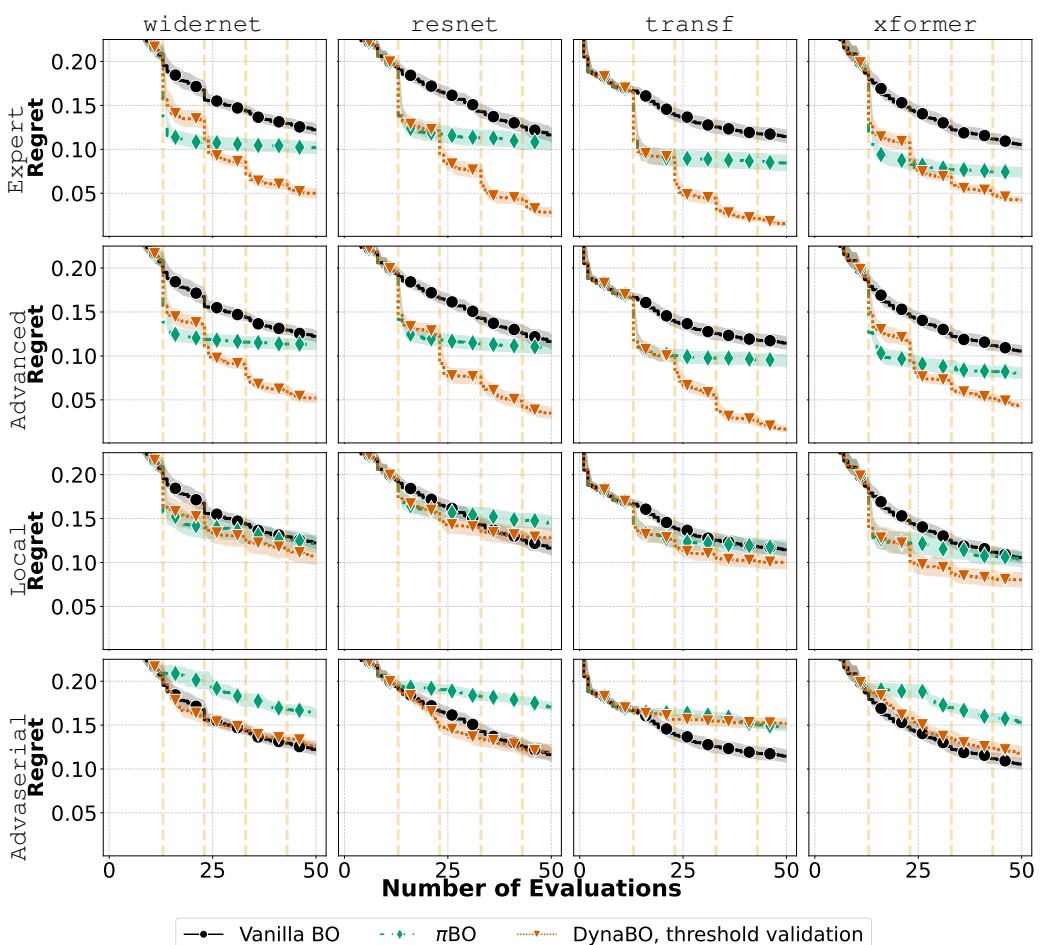

Figure 8: Mean regret for `PD1` using Expert, Advanced, Local, and Adversarial priors, with random forests as surrogate models. Priors are provided at vertical lines. The shaded areas visualize the standard error. The results indicate `DynaBO` outperforming $\pi$BO and remaining competitive with vanilla BO for adversarial priors.

### D.3 RANDOM PRIOR LOCATION

For experiments with randomly chosen prior locations, we model user behavior as follows. Each user provides an initial prior at the start of the optimization. If the last prior was given at time $t_i$, a new prior is provided at time $m$ with probability:

$$\mathbb{P}_{t_i}(\pi^{(m)}) = 1 - e^{-0.15 \cdot (m - t_i)}.$$

The resulting outcomes exhibit trends consistent with those observed in the main experiments.

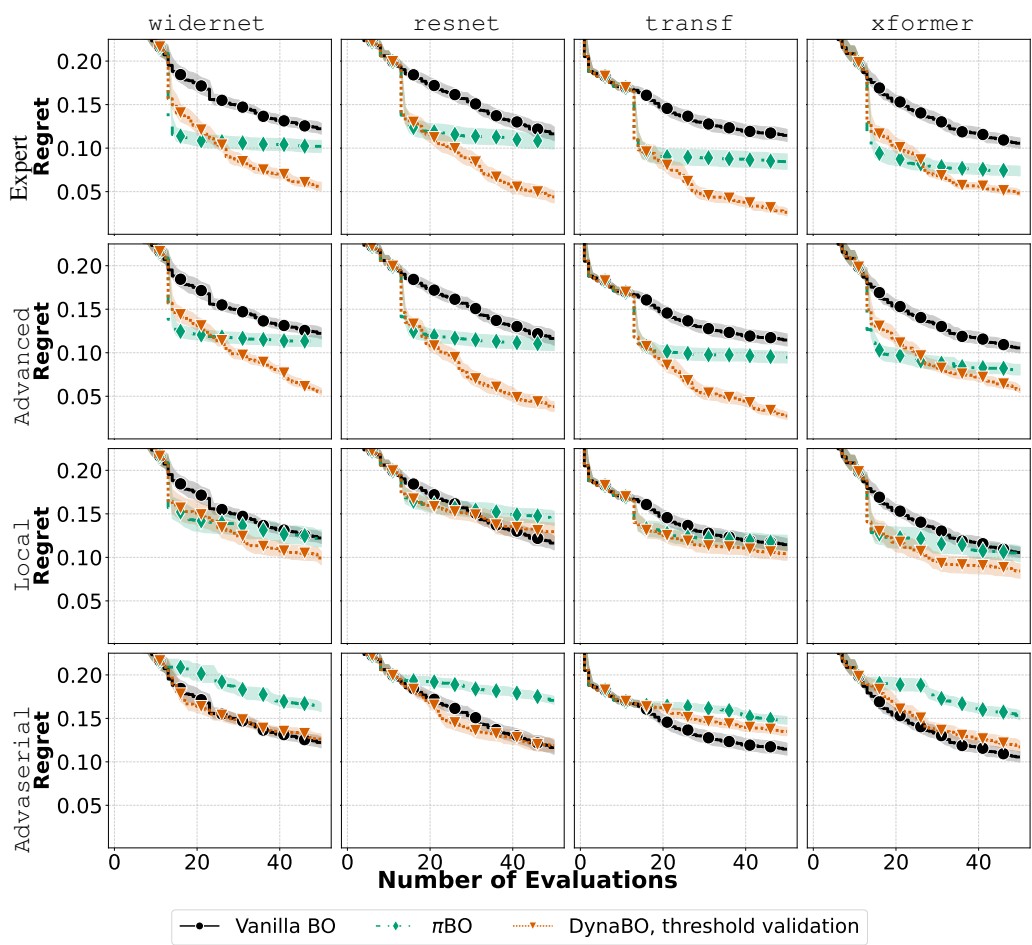

Figure 9: Mean regret for PD1 using Expert, Advanced, Local, and Adversarial priors, with random forests as surrogate models. The shaded areas visualize the standard error. The results indicate DynaBO outperforming $\pi$BO and remaining competitive with vanilla BO for adversarial priors.

## D.4 INCREASED BUDGET RESULTS

In our experiments with an increased budget, we introduce a prior every ten trials following the initial design. The indicate that random forests perform poorly without prior rejection, whereas Gaussian processes remain robust. Nevertheless, DynaBO with prior rejection consistently outperforms $\pi$BO on most scenarios.

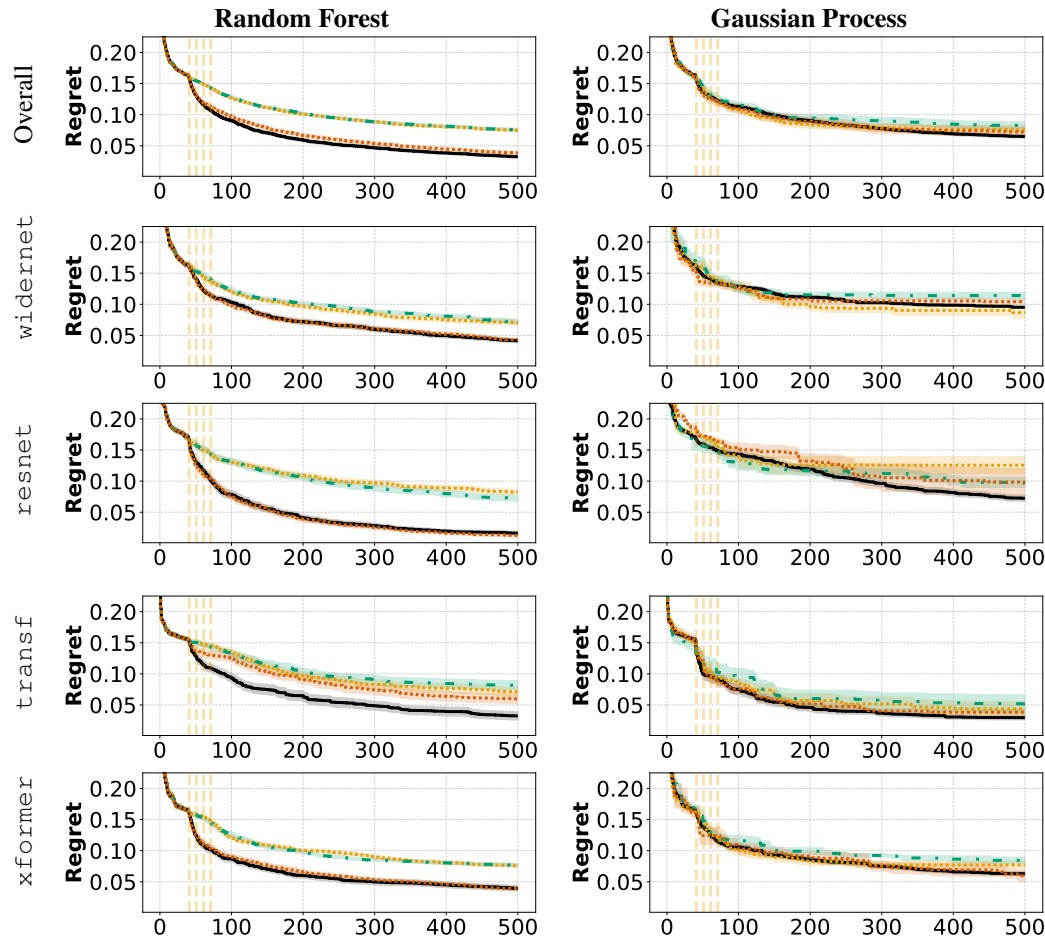

Table 2: Evaluation results across scenarios (rows) for two surrogate models (columns).

### D.5    NAÏVE DYNAMIC EXTENSION FOR $\pi$BO

In this ablation, we compare a less sophisticated baseline of naïvely removing old priors to the proposed mechanism of `DynaBO` for summing priors. The results are visualized in Figure 10. We find that no substantial performance difference can be observed between the two methods in our standard evaluation setup. This means that there is no harm in continuing to use the old priors.

However, to evaluate whether `DynaBO`'s intuition of old information being useful holds in practice, one has to evaluate the impact of comparing positive and negative priors. To that end, we evaluate what happens if an expert, advanced, or local prior, each provided with a chance of $1/3$ is followed by an adversarial prior. In this setup, the quality of the two methods differs significantly. For example, when positive priors are followed by negative priors, the results show a degraded performance, as can be seen in Figure 11. This result holds, even though the positive prior results in an immediate performance boost for both approaches.

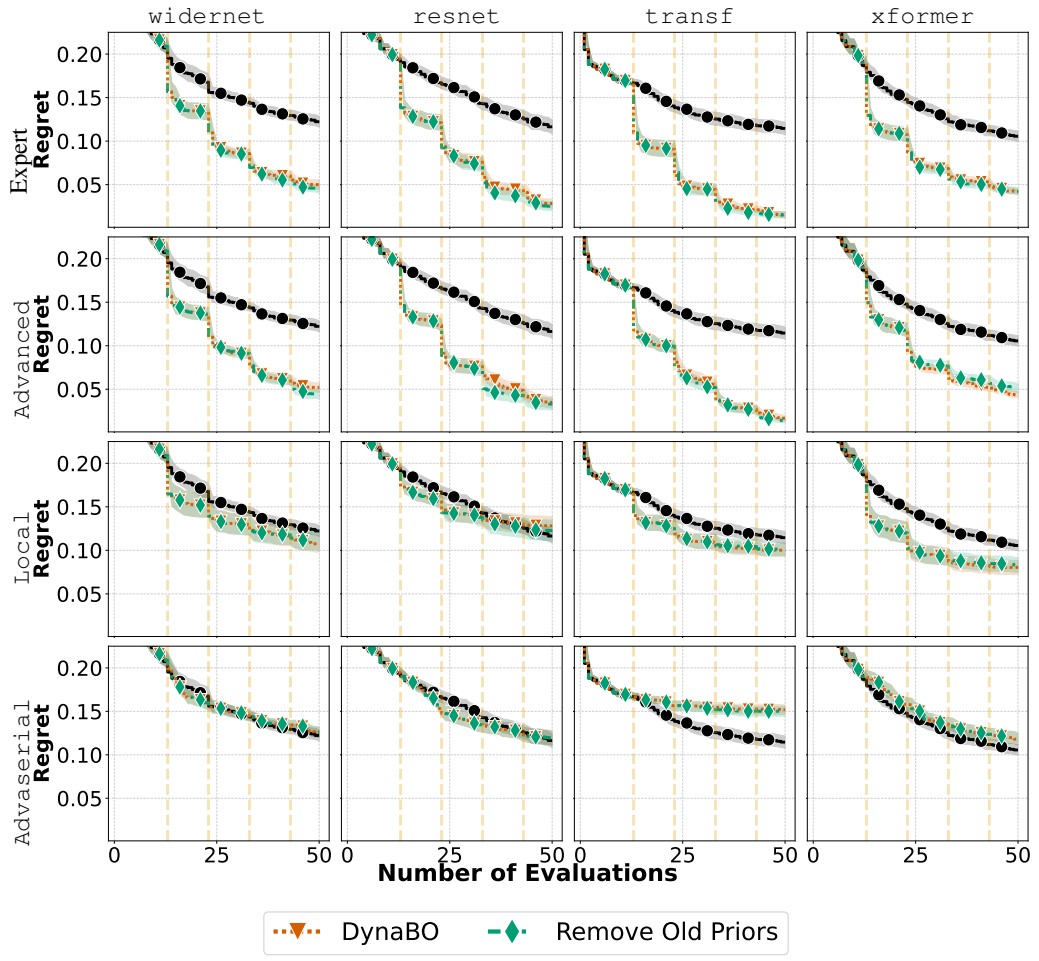

Figure 10: Mean regret for `PD1` using Expert, Advanced, Local, and Adversarial priors, with random forests as surrogate models. The shaded areas visualize the standard error. The results indicate `DynaBO` outperforming $\pi$BO and remaining competitive with vanilla BO for adversarial priors.

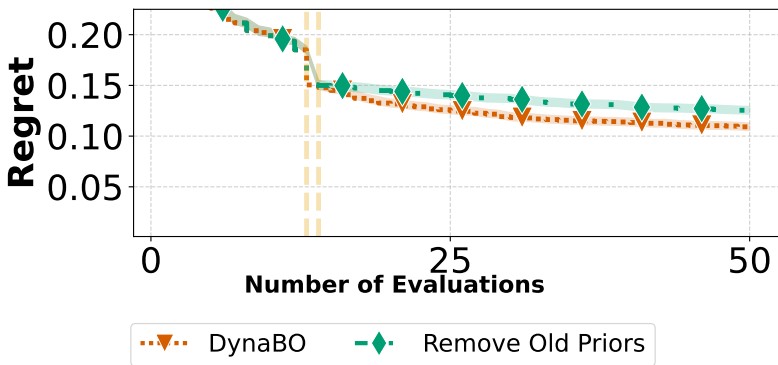

Figure 11: Investigation of helpful priors followed by adversarial priors.

### D.6   SENSITIVITY ANALYSIS OF THE PRIOR REJECTION CRITERION

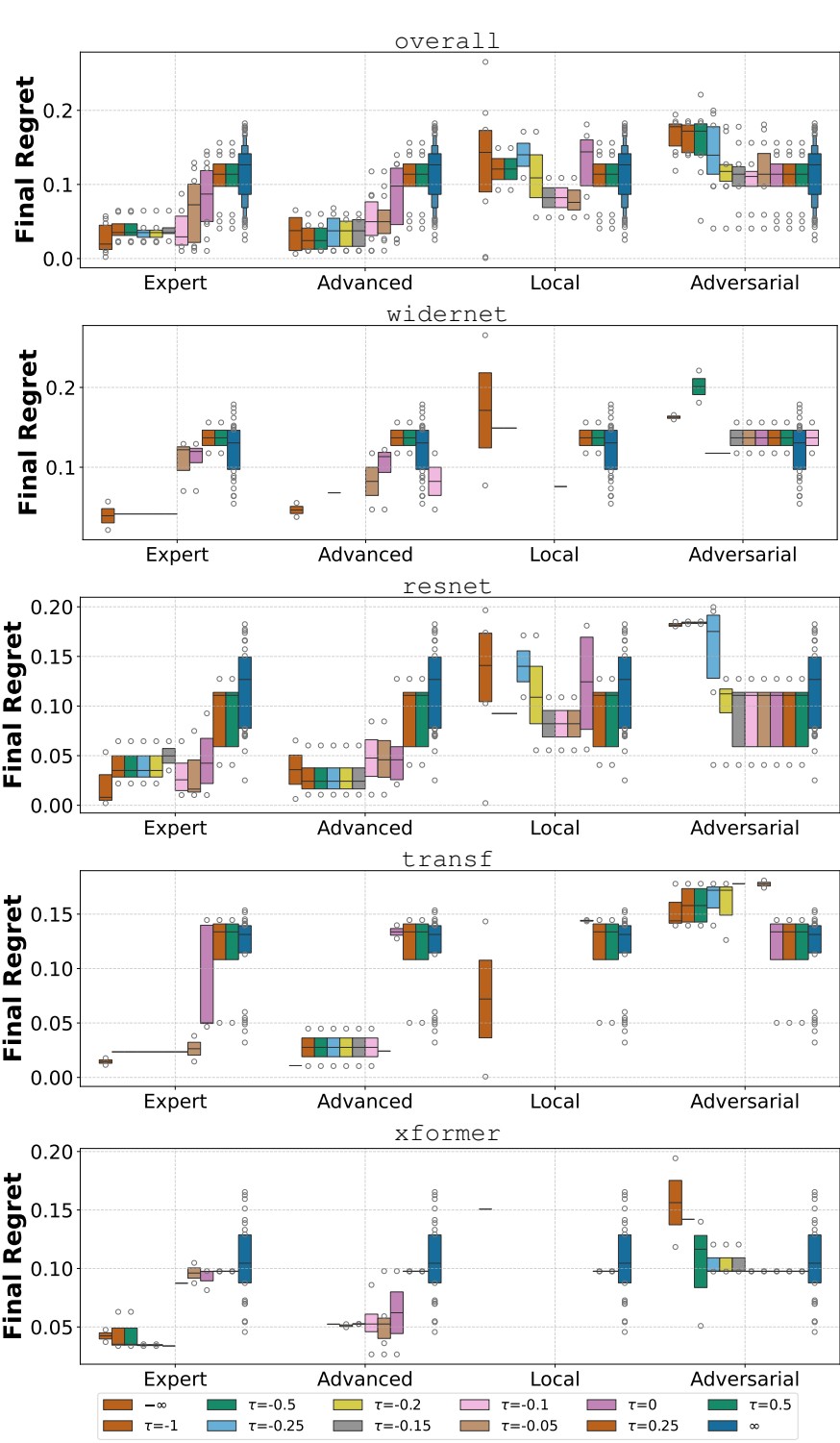

Figure 12: Sensitivity analysis of different thresholds $\tau$. $\tau = -\infty$ accepts all, and $\tau = \infty$ rejects all priors. `overall` contains the merged results from all scenarios.

### D.7 PRIOR DECAY ABLATION

Due to providing multiple priors during the experiment, we reinvestigated the speed at which priors are decayed. We therefore ablated the function $\phi$.

$$\alpha_{\hat{f}}^{\text{dyna}}(\lambda) := \alpha_{\hat{f}}(\lambda) \cdot \sum_{m=1}^{M} \pi^{(m)}(\lambda)^{\beta/\phi(t-t^{(m)})}$$

Our investigation reveals that the optimal decay rate varies according to the prior quality. In our paper, we utilize a linear decay, as it is a good trade-off between good and adversarial priors.

Table 3: Mean regret ($\mu$) and standard error (SE) for each prior type across decay configurations. All values are rounded to three decimal places.

| Config | Expert | | Advanced | | Local | | Adversarial | |
|---|---|---|---|---|---|---|---|---|
| | $\mu$ | SE | $\mu$ | SE | $\mu$ | SE | $\mu$ | SE |
| Logarithmic Decay | 0.027 | 0.001 | 0.033 | 0.001 | 0.105 | 0.002 | 0.177 | 0.001 |
| Linear Decay | 0.028 | 0.001 | 0.036 | 0.001 | 0.102 | 0.002 | 0.162 | 0.001 |
| Quadratic Decay | 0.034 | 0.001 | 0.041 | 0.001 | 0.103 | 0.002 | 0.149 | 0.001 |
| Cubic Decay | 0.038 | 0.001 | 0.042 | 0.001 | 0.104 | 0.002 | 0.146 | 0.001 |
| To the Power of 4 Decay | 0.039 | 0.001 | 0.046 | 0.001 | 0.101 | 0.002 | 0.145 | 0.001 |
| To the Power of 5 Decay | 0.039 | 0.001 | 0.045 | 0.001 | 0.103 | 0.001 | 0.145 | 0.001 |

# E  DECLARATION OF LLM USAGE

Throughout this submission, we made limited use of Large Language Models (LLMs) in the following ways:

- Code generation from specific instructions, primarily for producing plots and tables.
- Writing support, including translation and alternative phrasings.
- Assistance in locating related research.

All conceptual contributions, methodological developments, experimental designs, and analyses were carried out solely by the authors.

