# OpenReview forum: "Dynamic Priors in Bayesian Optimization for Hyperparameter Optimization"
_ICLR.cc/2026/Conference — Submitted to ICLR 2026_

### Official Review · Reviewer_CXds · 2025-10-28

**Soundness:** 3
**Presentation:** 3
**Contribution:** 2
**Rating:** 4
**Confidence:** 4

**Summary:**

The paper presents a Bayesian optimization approach that enables the incorporation of human-defined priors into the decision-making process. In contrast to previous work that only supported a single prior, this method allows for the use of multiple priors at different time steps.
To mitigate the risk of getting trapped in a local mode due to misspecified priors, the paper introduces a simple heuristic for rejecting priors when necessary.

**Strengths:**

- The paper is generally well written and easy to follow.

- The empirical evaluation appears thorough, comparing different priors across various benchmarks. The chosen baselines are reasonable, though the evaluation could potentially be enriched by including additional methods from the literature, for example Seng et al.

- The paper includes an ablation study on the sensitivity of the hyperparameter tau, which seems to play a central role in the proposed method.

**Weaknesses:**

Significance: While the motivation to incorporate user-provided priors into the optimization process is clear and conceptually appealing, I have some concerns regarding its practical applicability. In many real-world scenarios, I would argue that users might find it more intuitive to provide such priors only at the beginning of the optimization rather than continuously throughout the process. Especially in an AutoML context, where the goal is typically to maximize automation.

Moreover, the current procedure for constructing priors appears somewhat artificial and may not fully capture how practitioners would naturally define priors in practice. Finally, the proposed prior safeguard introduces an additional hyperparameter, τ, which, based on Figure 6, seems to have a substantial influence on performance and may need to be tuned according to the strength of the prior. This raises questions about the stability of the method in practical settings and whether it might introduce additional complexity in real-world applications.

Novelty: The main contribution, compared to the approach by Hvarfner et al., appears to be the weighting of the acquisition function based on the product of multiple priors instead of a single one. However, it seems that the method by Hvarfner et al. could, in principle, also replace the prior during the optimization process. Therefore, it remains somewhat unclear how much additional benefit is gained from using a product of multiple priors compared to relying on a single prior.

**Questions:**

- How are priors selected during the optimization process? For instance, in Figure 4, for expert priors, do you sample a new cluster at each decision point (vertical line)?

- Section 4.2, line 542: Could you clarify how a Normal distribution is placed on a prior?

- line 266: Do you mean f(\hat{\lambda}) - f(\lambda_{\star})

- Why do you use LCB for Equation 3 but EI as the acquisition function?



## Typos:

- line 327: missing \cite command

---

> ### Author Response · Authors · 2025-11-21
>
> Thank you very much for your thoughtful and positive feedback regarding clarity, empirical evaluation, and our ablation study on the sensitivity of the hyperparameter $\tau$. We address your questions and remarks below.
>
> # Remarks
>
> > Are continuous priors practical for practitioners?
>
> We agree that many ML practitioners provide priors upfront in fully automated setups. However, several works (e.g., Kannengießer et al., 2025) show that experts often avoid AutoML precisely because they seek more control during the process. Dynamic priors support exploratory workflows where users iteratively refine their intuition or information needs, not only final performance. Furthermore, HPO processes can be quite time-consuming, providing ample opportunity for a practitioner to intervene. If we allow for dynamic interactions, the practitioner can do so without actually interrupting the running HPO process.
>
> > Artificial prior construction
>
> While our priors are generated synthetically for controlled evaluation, we argue that pointing to specific configurations together with some confidence (as references for the probability distribution of the prior) is doable for users in practice. Crucially, DynaBO accepts any prior expressible as a probability distribution, including those derived from simple user actions, such as highlighting promising hyperparameter subranges. This keeps the method broadly usable and independent of how user priors are provided.
>
> > Should $\tau$ be tuned according to the prior belief, thereby introducing complexity?
>
> The introduced parameter, $\tau$, does not scale with prior strength; confidence in the prior is expressed via the prior’s standard deviation. $\tau$ solely controls how aggressively the safeguard rejects potentially unhelpful priors. Across all experiments, we use a consistent $\tau=-0.15$, demonstrating robustness.
>
> > Main contribution appears to be only multiplication of priors
>
> We respectfully disagree with the notion of the multiplication of priors being the only contribution of our work and would like to highlight that our contributions are twofold:
> First, we introduce a generalization that enables the approach's users to provide multiple priors during the optimization process, with a theoretical convergence guarantee despite misleading priors and a speedup guarantee in case of usefulness.
> And, second, we introduce a safeguard mechanism that can efficiently detect and protect against misleading priors by utilizing the surrogate model’s belief in optimization potential.
> Lastly, while not a contribution per se, we introduce a principled prior construction scheme for our evaluation and extend the theoretical framework of $\pi$BO to the multi-prior setting.
>
> > Suggested baseline: replacing the prior at each step
>
> Thank you for this helpful suggestion. We conducted further experiments for this approach as a baseline to our experiments. The new results can be found in Appendix D.5. In sumarry, we make the following observations:
> If all priors provided to the approach are of the same kind, both the baseline and DynaBO perform on a par.
> With mixed-quality priors (e.g., expert -> adversarial), replacing priors also replaces helpful information. This highlights the main benefit of DynaBO, which retains the beneficial parts
> The full results are shown in Appendix D.5.
>
> # Questions
> > Q1: How are priors sampled?
>
> As described in Appendix B.1, at each prior point, we sample clusters of priors based on the incumbent and the respective prior policy. Potentially, a cluster can also be sampled multiple times if the cluster is not excluded due to the incumbent’s performance.
>
> > Q2: Could you clarify how a Normal distribution is placed on a prior?
>
> Appendix C.3 describes this, and we now explicitly reference Equation (4) in Appendix B.1 for clarity. The configuration that is sampled from a cluster is used as the mean of the distribution. For continuous hyperparameters, the normal distribution is placed as usual. For categorical hyperparameters a Dirac distribution is used allocating a probability mass of 1-$\epsilon$ on the value set in the configuration, other values are assigned a very small value, summing up to $\epsilon$.
>
> > Q3: Do you mean $f(\hat{\lambda}) - f(\lambda_{\ast})?$
>
> Thank you - corrected.
>
> > Q4: Why LCB for Equation 3 when EI is used for optimization?
>
> We think that LCB represents a valid choice for our experiments, as it estimates the optimistic potential of unexplored prior regions, especially those with high uncertainty. This behavior better reflects the goal of the safeguard against unhelpful priors, i.e., estimating the tuning potential of a given prior and only rejecting a prior if there is no such potential. The acquisition function used for optimization can differ; we used EI in our experiments, as this represents a common choice when performing HPO via BO.
>
> ## References
> Kannengiesser et al. 2025, Practitioner Motives to Use Different Hyperparameter Optimization Methods

---

### Official Review · Reviewer_qADC · 2025-10-31

**Soundness:** 1
**Presentation:** 2
**Contribution:** 2
**Rating:** 2
**Confidence:** 4

**Summary:**

The paper considers the problem of informing hyperparameter optimization (HPO) via Bayesian Optimization through a user-specified “priors” / priority functions, which are used to reweigh the BO acquisition function.  Expanding on previous work which introduced this idea, the proposed approach uses multiple priority functions introduced at different time points, which are aggregated multiplicatively.  An additional filtering step is used to reject potentially priority functions with outsized impact.  A theoretical analysis aims to provide convergence properties (Thm 1), robustness to misleading priors (Thm 2) and utility of informative priors (Thm 3). An experimental study evaluates the proposed method on heuristically defined, data-driven priors (obtained using BO also).

**Strengths:**

- Putting the human in the loop with HPO is an interesting (but not new) problem.
- The experimental study is carried out on realistic HPO tuning tasks (albeit I have concerns about the choice of priors as below).

**Weaknesses:**

- The theoretical analyses provided are only asymptotic in nature; this is especially misleading for Theorem 2, where for finite-time, clearly misspecified priors will slow down performance.  The analysis essentially relies on the fact that asymptotically the effect of the prior vanishes / becomes trivial.
- The theoretical analysis seems to have some issues, and the exposition contains some vague / unjustified statements (see questions below)
- While the objective functions in the experiment come from realistic AutoML tasks, the choice of priors seems rather simplistic (Section 6.1)
- In contrast to prior work on human-in-the-loop HPO (cf Xu et al ‘24), the paper does not actually carry out any experiments with humans in the loop.

**Questions:**

- It seems rather odd that for a Bayesian method, the “prior” is not encoded as a Bayesian prior (e.g., through choice of the mean / kernel function).  Why was this not considered?
- There seems to be an implicit assumption that acquisition function needs to be positive (such as expected improvement / probability of improvement). This is generally not the case, e.g. for UCB/LCB.  Is any normalization etc. assumed?
- The approach also seems to require that the design space is continuous.  What about discrete design spaces? The filtering rule (3) would not apply?
- How should the parameters of the normal distribution in (3) to be picked? What about the threshold \tau?
- Thm 1: what are assumptions on acquisition function? The approach is stated in generality, but the proof of Theorem 1 seems to implicitly rely on using UCB?
- There seems to be an issue in the analysis:  Page 16 states that sublinear regret (i.e.., 1/T\sum_{t=1}^T (f(\lambda*)-f(\lambda_t)) \to 0 for t\to \infty) implies vanishing instantaneous regret (i.e., f(\lambda_t)\to f(\lambda*)) – this is not generally true!  Are there any additional implicit assumptions?
- Some statements are quite vague.  E.g., A.1: “established independently of the UCB acquisition function” very vague; it doesn’t hold e.g., for trivial acquisition functions that only pick a single action. Again, are there any implicit assumptions?
- The statement “asymptotic convergence rate improves” in A.3 also not clear. Can you please elaborate?

---

> ### Author Response · Authors · 2025-11-21
>
> We thank the reviewer for highlighting the importance of our work on keeping the human in the loop with HPO approaches, and for engaging deeply with the theoretical aspects of our work. Below, we address the concerns and questions raised.
>
> # Concerns
> > Asymptotic nature of analysis
>
> We agree that considering a finite-time horizon a misleading prior will slow down the optimization process. What we aim to show in Theorem 2 is that it does not affect the asymptotic behavior and convergence guarantees of BO. As those guarantees are asymptotic in nature, so are ours. To our knowledge, fixed-time results are not conceivable without strong assumptions on the optimization task; if the reviewer has pointers, we would greatly appreciate them.
>
> # Questions
> > Q1: Why not define priors via GP mean/kernel?
>
> We agree that your proposed priors are more Bayesian conceptually. However:
> Specifying meaningful mean/kernel functions requires deep BO/GP expertise.
> In our approach, users can indicate regions in the configuration space, requiring less expertise in BO
> Kernel or mean modification changes the surrogate model itself, making the effect more opaque.
> We argue that users should not need to become experts in BO/GPs, but rather leverage observations or knowledge available to them (i.e., what are good HP settings for previous or related tasks).
>
> > Q2: Positivity of acquisition functions
>
> We note that, under the common practice of normalizing observations, which is also beneficial for GP hyperparameter tuning, standard acquisition functions are non-negative. Among widely used acquisition functions, UCB/LCB are the primary exceptions that can produce negative values in certain cases; for these, simple clipping can be used for handling these exceptions. We have added this clarification to the manuscript in Section 2.
>
> > Q3: Does the method require continuous spaces?
>
> In fact, our method does apply to continuous, discrete, and mixed spaces. The xgboost benchmark (10 continuous, 2 integer, 2 categorical) already demonstrates this. In our prior rejection mechanism, we sample from the prior distribution (as we describe in Appendix B.3 of the revised manuscript). For comparison to the incumbent region, we fix categorical hyperparameters to the value of the incumbent and sample the remaining hyperparameters according to a normal distribution.
>
> > Q4: How to pick parameters of the normal distribution and $\tau$?
>
> Appendix B.1 discusses the prior construction, and Appendix C.3 describes our implementation of prior rejection in detail.
> For the mean of the prior’s normal distribution, we take the prior configuration itself.
> The standard deviation reflects how specific or vague the user intends the prior to be.
> $\tau$ controls the safeguard’s permissiveness; throughout all experiments (except the $\tau$-ablation), we consistently use $\tau=-0.15$, which works robustly across settings.
>
> > Q5: Assumptions on the acquisition function in Thm 1
>
> For theory, we explicitly assume UCB/LCB, as stated at the beginning of Appendix A. Empirically, DynaBO also works well with EI, as we show in Figure 4 in Section 6.3.
>
> > Q6: Clarification on sublinear vs instantaneous regret
>
> Thank you for catching this oversight on our side. Indeed, the sublinear cumulative regret does not imply vanishing instantaneous regret. We now clarified in Appendix A that our almost-sure convergence proof relies on UCB’s known almost-sure convergence property, not the regret bound, and that the prior adjustment of the acquisition function does not interfere with that.
>
> > Q7: Vague statement in A.1
>
> Thank you for pointing this out. We have now clarified in Appendix A.1 that our guarantees require an acquisition function with almost-sure convergence (as is the case with UCB/LCB), and we have adjusted the exposition accordingly.
>
> > Q8: “Asymptotic convergence rate improves”
>
> We clarified the wording and strengthened the argument in Appendix A.3: Theorem 3 shows that the cumulative regret bound improves relative to standard BO when priors concentrate near the optimum.

---

### Official Review · Reviewer_Tn66 · 2025-11-01

**Soundness:** 3
**Presentation:** 3
**Contribution:** 3
**Rating:** 6
**Confidence:** 4

**Summary:**

The paper proposed DynaBO, a Bayesian Optimization algorithm that can integrate prior information into the standard BO framework to improve the solutions. DynaBO generalizes piBO, a state-of-the-art prior-based BO algorithm, from a single to multiple priors provided by users over time. The key idea is to take the product of the user-specified priors and multiply by the acquisition function values, generating a dynamically-adapted acquisition function, which are maximized for the next data points. The authors further propose to reduce the impact of older priors by scaling them with the inverse of the time step factor. The proposed method is supported by a theoretical analysis of the convergence rate, robustness to misleading priors and convergence acceleration with informative priors. Empirically, DynaBO is compared with vanilla BO and piBO on many benchmark problems, under different prior settings (expert, advanced, local and adversarial).

**Strengths:**

- The paper is clearly written.
- The idea is novel, extending a previous work to incorporate dynamic priors.
- There are many analyses to support the work, including theoretical analysis and experimental results. Regarding the experiments, there are many different benchmark settings, including different prior settings (from informative to misleading priors), different prior supplying times (fixed and random times) and different surrogate models (Gaussian Process and Random Forest).

**Weaknesses:**

1.	The choice of square exponent in the formula in line 206 is not clearly explained. The authors mention it is to avoid the slow fading of old priors, but it still does not explain why square (rather than other functions such as cubic, etc.) should be used. There should be another ablation study on this choice, because I think the fading speed of older priors is an important factor to consider.
2.  It is not clear why some baselines are not compared in the experiments, such as PriorBand (Mallik et al. 2023) or BOPrO (Souza et al., 2021a).

**Questions:**

1.	Would there be any issue if more priors are supplied? By definition, the equation in line 206 computes the product of the priors, so when there are many (but still finite) priors, the product would become nearly 0 regardless of the original acquisition values, because each prior is in (0, 1]. According to Appendix B.2, some clippings are introduced to keep the priors at 1e-12, however, given a large number of priors, there will be many priors with very small values, which I think will still reduce the final product to be very small, and affect the original acquisition function values. Can the authors elaborate more on this? Also, have the authors run experiments with more priors in increased budget scenarios (Fig. 5), such as with 20-50 priors?
2.	Regarding the prior rejection threshold tau, in line 264, the authors explained about the choices of tau for EI and LCB acquisition functions. How about other acquisition function choices? Also, will the ablation study in Sec. 6.4 still be correct if EI and LCB are not used?

---

> ### Author Response · Authors · 2025-11-21
>
> Thank you very much for your thoughtful and positive feedback regarding clarity, novelty, theoretical analysis, and evaluation. We address your remarks and questions below.
>
> # Remarks
> > Choice of square exponent & ablation
>
> Thank you for highlighting the additional insights that could be gained from an ablation study. The submitted paper version indeed lacked an ablation. With the updated acquisition function (i.e., using the sum of priors rather than the product), the quadratic decay actually no longer provides the benefit it did before. However, to demonstrate this empirically, we added a hyperparameter sensitivity study. This can also be found in Appendix D7.
>
> | Config                 | Expert          | Advanced       | Local    | Adversarial     |
> |------------------------|---------------|---------------|----------------|----------------|
> | Logarithmic Decay      | 0.027 ± 0.001 | 0.033 ± 0.001 | 0.105 ± 0.002  | 0.177 ± 0.001  |
> | Linear Decay           | 0.028 ± 0.001 | 0.036 ± 0.001 | 0.102 ± 0.002  | 0.162 ± 0.001  |
> | Quadratic Decay        | 0.034 ± 0.001 | 0.041 ± 0.001 | 0.103 ± 0.002  | 0.149 ± 0.001  |
> | Cubic Decay            | 0.038 ± 0.001 | 0.042 ± 0.001 | 0.104 ± 0.002  | 0.146 ± 0.001  |
> | To the Power of 4 Decay| 0.039 ± 0.001 | 0.046 ± 0.001 | 0.101 ± 0.002  | 0.145 ± 0.001  |
> | To the Power of 5 Decay| 0.039 ± 0.001 | 0.045 ± 0.001 | 0.103 ± 0.001  | 0.145 ± 0.001  |
>
> > Missing baselines (PriorBand, BOPrO)
>
> As PriorBand and BOPrO are discussed in the related work, arguably, one would also expect them as baselines. However, in our experiments, we do not use multi-fidelity evaluations, which excludes PriorBand as a baseline. In turn, BOPrO is substantially outperformed by $\pi$BO (Hvarfner et al., 2022). To maintain a more focused comparison, we thus focused on $\pi$BO as the most competitive baseline for our problem.
> We have updated the text in Section 6.2 to explicitly justify these choices.
>
> # Questions
> > Q1: Behavior with many priors
>
> Thank you for this important question. In the previous formulation (multiplying priors), many priors could indeed cause the acquisition values to collapse toward zero. This could have led to priors hindering the optimization process due to a uniform acquisition function. We resolved this by switching to a sum of individually decaying priors in the acquisition function (see our general comment above and the updated Section 4.1):
> $$ \alpha^\text{dyna}_{\hat{f}}(\lambda)$$
>
> $$= \alpha_{\hat{f}}(\lambda) \cdot (\frac{1}{M} \sum_{m=1}^{M} \Pi^{(m)}(\lambda)^{\beta/(t-t^{(m)})^2}) $$
> This updated acquisition function prevents the collapse of the acquisition values and ensures a stable behavior even with many user interventions.
>
> > Q2: Does the $\tau$-ablation still hold with other acquisition functions?
>
> In our approach, we proposed a general framework to reject priors grounded in the belief learned by the surrogate model. Theoretically, the user can replace the acquisition function as needed. We believe that the utilization of LCB is uniquely well-equipped in this setting, as it focuses on the potential of regions rather than assessing them greedily. We adapted our paper to make this clearer. Nevertheless, you are correct that replacing our proposed acquisition function requires choosing $\tau$ properly.
>
> ## References
> Hvarfner, C., Stoll, D., Souza, A., Lindauer, M., Hutter, F., & Nardi, L. (2022). πBO: Augmenting Acquisition Functions with User Beliefs for Bayesian Optimization. In Tenth International Conference of Learning Representations, ICLR 2022.

---

### Author Response · Authors · 2025-11-21

# General Comment on the Paper Adaptations
We sincerely thank all reviewers for their detailed and constructive feedback. We are pleased that the clarity of the paper, the novelty of incorporating dynamic priors into Bayesian optimization, and the breadth of our theoretical and empirical analyses were positively received. We also appreciate the recognition of the practical relevance of human-in-the-loop HPO. We are grateful for the suggestions on how to further strengthen the work, and we have incorporated several improvements accordingly.

Following the reviewers’ feedback, in particular Reviewer Tn66’s comments, we modified the acquisition function in Section 4.1: rather than multiplying all priors, we now sum them before combining them with the acquisition function, while still decaying each prior individually. This avoids the degeneration previously caused by multiplying many priors and removes the need for a quadratic decay. We have updated all theoretical results accordingly and explicitly demonstrate that they remain valid under this change. We additionally re-executed all experiments and found no significant change in the results.

We additionally revised the manuscript for clarity throughout. Changes are marked by blue text. If a figure has been replaced or updated, the caption is marked in blue.

---

### Author Response · Authors · 2025-12-03
**Revisions Incorporate Feedback, Strengthening Clarity, Theory, and Experiments**

We thank the reviewers and area chairs for their time investment and their thoughtful feedback. While the discussion period was disrupted, the comments we received were clear and easily actionable. We have incorporated the suggestions in a focused manner without altering the core method or introducing new assumptions.

We respectfully request that the Area Chair considers our rebuttal in their final decision. The reviewers had already highlighted the novelty, theoretical analysis, clarity, and overall evaluation setup. The average score, prior to the rebuttal, is near the center of this year’s score distribution (see https://papercopilot.com/statistics/iclr-statistics/iclr-2026-statistics/). However, the limited ratings were tied to specific concerns (clarity, theoretical exposition, and comparative evaluation) that have now been addressed in the revision. With these issues resolved, we hope that the paper is considered in light of its present clarity, rigor, and strengthened empirical support.

**Reviewer Tn66:** We clarified the acquisition-function adaptation and added a targeted ablation on the prior-decay rate. These refinements address the reviewer’s concerns while preserving the original general design and conclusion.

**Reviewer qADC:** We improved the presentation of the theoretical analysis by reorganizing proofs, clarifying assumptions, and adding intermediate explanations. In particular, we revised a part of the theoretical analysis to provide a rigorous proof of the convergence properties in compliance with the reviewer’s remarks. These revisions increase readability and rigor without affecting the substance or validity of the results.

**Reviewer CXds:** We expanded the empirical section with an additional baseline comparison (Appendix D.5), showing that DynaBO consistently outperforms a naive PiBO extension under priors of varying quality (similar to the proposal by Seng et al, without their special surrogate model). This directly supports our claims regarding the method’s practical advantages and highlights the necessity of DynaBO’s methodology.

Overall, the revisions primarily strengthen exposition and empirical evidence. We believe the updated manuscript reflects the reviewers’ feedback carefully and is now presented in a clear and complete form.

---

### Meta-Review · Area_Chair_1uH4 · 2026-01-05

**Summary:**

This paper presents a method for steering Bayesian optimization in-the-loop during the optimization process by allowing users to specify knowledge / preferences as "priors".  The reviews were mixed (6, 2, 4) but leaning towards reject with two marginal scores and one reject.  Some reviewers noted that the paper was well written, easy to follow and interesting.  Reviewers raised issues regarding novelty, finding the work somewhat incremental over recent related work.  They also had concerns regarding how realistic the priors / experiments were.  The reviewer recommending reject voiced significant concerns regarding the theoretical guarantees of the method - i.e. that the theory is based on asymptotic convergence but "The analysis essentially relies on the fact that asymptotically the effect of the prior vanishes / becomes trivial.".  This does raise questions about how meaningful the theory is in regard to the paper's contribution (i.e. incorporating priors to improve convergence rate).  The same reviewer also raised concerns regarding various imprecise or unjustified statements in the paper.

**Reviewer Concerns:**

The reviewer proposing a reject (score of 2) raises reasonable concerns regarding the technical contributions of the paper.  Unfortunately, it would require a very significant modifications / improvement to change this score to an accept.  The authors don't really address this reviewer's concerns.  E.g., the reviewers acknowledge the challenges with asymptotic theoretical analysis, but don't change their approach.  It does seem as if the theory is present in the paper in order to make the technical contribution appear more significant, without really saying anything about the method in this paper - i.e. what can you say about the rate of convergence?  I agree that the statement "Asymptotically, the algorithm does not suffer degradation in performance even when faced with misleading priors" is good to know, but somewhat vacuous given the "asymptotically" since the prior should vanish in the limit.

While the authors addressed reviewers comments point-by-point, most of the authors' responses seemed superficial, without really changing the substance of the paper.

**Reviewer Scores:**

I don't think the reviewers would have changed their scores substantially (or substantially enough to make a difference).

---

### Decision · Program_Chairs · 2026-01-26

Reject